# Consistent Sufficient Explanations and Minimal Local Rules for explaining any classifier or regressor

**Salim I. Amoukou**
LaMME
University Paris Saclay
Stellantis Paris

**Nicolas J-B. Brunel**
LaMME
ENSIIE, University Paris Saclay
Quantmetry Paris

## Abstract

To explain the decision of any regression and classification model, we extend the notion of probabilistic sufficient explanations (P-SE). For each instance, this approach selects the minimal subset of features that is sufficient to yield the same prediction with high probability, while removing other features. The crux of P-SE is to compute the conditional probability of maintaining the same prediction. Therefore, we introduce an accurate and fast estimator of this probability via random Forests for any data $(\boldsymbol{X}, Y)$ and show its efficiency through a theoretical analysis of its consistency. As a consequence, we extend the P-SE to regression problems. In addition, we deal with non-discrete features, without learning the distribution of $\boldsymbol{X}$ nor having the model for making predictions. Finally, we introduce local rule-based explanations for regression/classification based on the P-SE and compare our approaches w.r.t other explainable AI methods. These methods are available as a Python package[1].

## 1 Introduction

Many methods have been proposed to explain specific predictions of machine learning models from different perspectives, such as feature attributions approaches [Lundberg and Lee, 2017, Ribeiro et al., 2016], decision rules [Ribeiro et al., 2018], counterfactual examples [Wachter et al., 2017] and logic-based [Shih et al., 2018, Darwiche and Hirth, 2020].

Among these categories, the most popular are feature attributions approaches, in particular SHAP [Lundberg and Lee, 2017], which is based on Shapley Values (SV) and aims at indicating the importance of each feature in the decision. One of the main reasons for SHAP's success is its scalability, nice representations of the explanations, and mathematical foundations. However, SV used in SHAP does not guarantee the truthfulness of the important variables involved in a given decision. Indeed, it is possible to construct simple theoretical models (defined on a partition of the feature space) for which SV cannot distinguish between local important and non-important variables (see conclusion in Amoukou et al. [2022]). Similar difficulties have also been highlighted by Ghalebikesabi et al. [2021] for SHAP and LIME [Ribeiro et al., 2016]. This lack of guarantees is a major issue since the explanations may be used for high-stakes decisions. Moreover, Additive Explanations are not suitable when interactions occur in the model [Gosiewska and Biecek, 2019].

An appealing solution to the problem above is to use decision rules [Ribeiro et al., 2018] or logic-based explanations [Darwiche and Hirth, 2020, Shih et al., 2018] which gives local explanations that take into account interactions while ensuring minimality and guarantee on the outcome. However, these methods are not currently available in the general case (e.g., regression model, continuous

---

[1] https://github.com/salimamoukou/acv00

36th Conference on Neural Information Processing Systems (NeurIPS 2022).

features, . . . ). Our objective is to extend these methods to more realistic cases by developing new consistent algorithms.

In this paper, we generalize the concept of Probabilistic Sufficient Explanations (P-SE) introduced by Wang et al. [2020]. P-SE is a relaxation of logic-based explanation: it explains the classification of an example by choosing a minimal subset of features guaranteeing that, the model makes the same prediction with high probability, whatever the values of the remaining features (under the data distribution). Such a subset is called a Sufficient Explanation (also known as sufficient reason or prime implicant [Shih et al., 2018, Darwiche and Hirth, 2020]).

We make several contributions. We extend the concept of Same Decision Probability (SDP) to the regression setting so that we can extend Sufficient Explanations from classification to regression. We introduce a fast and efficient estimator of the SDP based on Random Forests and prove its uniform almost sure convergence. Our approach can deal with non-discrete features and does not need the estimation of the distribution of $\boldsymbol{X}$, contrary to [Wang et al., 2020]. Our method can explain the data generating process $(\boldsymbol{X}, Y)$ directly or any learnt model $(\boldsymbol{X}, f(\boldsymbol{X}))$.
We introduce the probabilistic local explanatory importance which is the frequency of each feature to be in the set of all Sufficient Explanations. In particular, this indicates the diversity of the Sufficient Explanations. We introduce local rule-based explanations for classification or regression which are simultaneously minimal and sufficient. We compare our approaches w.r.t other explainable AI methods and provide a Python package that computes all our methods.

## 2 Motivations and Related works

The methods used to explain the local behavior of Machine Learning models can be organized into 5 groups: features attributions, decision rules, instance-wise feature selection, logical reasoning approaches, data generation based or counterfactual examples. The benefits of feature attribution-based explanations, e.g., SHAP [Lundberg et al., 2020] or LIME [Ribeiro et al., 2016] is that they are easy to read, they can be applied to any model and are generally more scalable than their alternatives. On the other hand, they are sensitive to perturbations [Ignatiev et al., 2019], or can be fooled by adversarial attacks [Slack et al., 2020]. These downsides can be caused by the local perturbations used, which make them inconsistent with the data distribution.

Quite differently, instance-wise feature selection such as L2X [Chen et al., 2018] or INVASE [Yoon et al., 2018] aims at finding the minimal subset of variables that are relevant for a given instance $\boldsymbol{x}$ and its label $y$. Interactions can be captured in that way. In addition, the identification of a minimal subset $S = S(\boldsymbol{x})$ is well-formalized and the objective is to find $S$ such that $\mathcal{L}(Y|\boldsymbol{X} = \boldsymbol{x}) \approx \mathcal{L}(Y|\boldsymbol{X}_S = \boldsymbol{x}_S)$. However, these methods are not reliable because they are prone to approximation errors due to the training of several Neural Networks, and they provide no guarantees about the fidelity of the explanations [Jethani et al., 2021]. A similar approach is also developed in [Dhurandhar et al., 2018] called Pertinent Positive.

Anchors [Ribeiro et al., 2018] are local rule-based explanations that propose a solution to the reliability issue by providing an explanation with guarantees. It explains individual predictions of any classification model by finding a decision rule that reaches a given accuracy for a high percentage of the neighborhood of the instance. However, the method is only available for classification, requires discrete variables, is unstable and tends to use more variables than needed.

Logical Reasoning Approaches such as Sufficient Reasons [Shih et al., 2018, Darwiche and Hirth, 2020] select a minimal subset of features guaranteeing that, no matter what is observed for the remaining features, the decision will stay the same. It can be seen as an instance-wise feature selection but with guarantees of sufficiency and minimality (i.e., no subset of the set satisfies the sufficiency condition). However, since the guarantees are deterministic, it is often necessary to include many features into the explanation, making the explanation more complex and thus less intelligible. A relaxation of this method is the Sufficient Explanations [Wang et al., 2020] that gives probabilistic guarantees instead of deterministic guarantees, i.e., it required that the prediction remains the same with high probability. It gives a simple local explanation with guarantees while considering feature interactions and the data distribution. However, it is limited to classification with binary features and requires learning the distribution of the features. Moreover, the Sufficient Explanations are not unique, which causes a selection problem as the whole set of explanations is not interpretable.

In this work, we propose a consistent method that efficiently finds the Sufficient Explanations of any data generating process $(\boldsymbol{X}, Y)$ or any model $(\boldsymbol{X}, f(\boldsymbol{X}))$, without learning the distribution of $\boldsymbol{X}$. In particular, we don't need to have access to the model $f$, we need only the predictions, contrary to [Wang et al., 2020]. We propose local attributions that summarize the diversity of the Sufficient Explanations. In addition, we propose local rule-based explanations for regression and classification models based on Sufficient Explanations. To the best of our knowledge, it is the first local rule-based explanations for regression.

## 3 Probabilistic Sufficient Explanations for Regression

Let assume we have an i.i.d sample $\mathcal{D}_n = (\boldsymbol{X}^i, Y^i)_{i=1,\dots,n}$ such that $(\boldsymbol{X}, Y) \sim P_{(\boldsymbol{X}, Y)}$ where $\boldsymbol{X} \in \mathbb{R}^p$ and $Y \in \mathbb{R}$. We use uppercase letters for random variables and lowercase letters for their value assignments. For a given subset $S \subset [p]$, $\boldsymbol{X}_S = (X_i)_{i \in S}$ denotes a subgroup of the features.

We define the explanations of an instance $\boldsymbol{x}$ as the minimal subsets $\boldsymbol{x}_S, S \subset [p]$ such that given only those features, the model yields "almost" the same prediction $y$ as on the complete example with high probability, under the data distribution $p(\boldsymbol{X})$. The main probabilistic reasoning tool that we use for our explanations is the Same Decision Probability (SDP) [Chen et al., 2012]. For classification, it is defined as the probability that the classifier has the same output by ignoring some variables. To explain also regression models, we propose the following definition of the SDP:

**Definition 3.1. (Same Decision Probability of a regressor).** Given an instance $(\boldsymbol{x}, y)$, the Same Decision Probability at level $t$ of the subset $S \subset [\![1, p]\!]$, w.r.t $\boldsymbol{x} = (\boldsymbol{x}_S, \boldsymbol{x}_{\bar{S}})$ is

$$SDP_S(y; \boldsymbol{x}, t) = P\left((Y - y)^2 \le t \,|\, \boldsymbol{X}_S = \boldsymbol{x}_S\right).$$

In a regression setting, the SDP gives the probability to stay close to the same prediction $y$ at level $t$, when we fix $\boldsymbol{X}_S = \boldsymbol{x}_S$ or when $\boldsymbol{X}_{\bar{S}}$ are missing. The higher is the probability, the better is the explanation powered by $S$. Note that for classification, the SDP is defined as $SDP_S(y; \boldsymbol{x}) = P(Y = y \,|\, \boldsymbol{X}_S = \boldsymbol{x}_S)$. Although we present all the methods with the SDP for regression, they remain the same for classification, we only need to replace $SDP_S(y; \boldsymbol{x}, t)$ by $SDP_S(y; \boldsymbol{x})$. Now, we focus on the minimal subset of features such that the model makes the same or almost the same decision with a given (high) probability $\pi$.

**Definition 3.2. (Minimal Sufficient Explanations).** Given an instance $(\boldsymbol{x}, y)$, $S_\pi(\boldsymbol{x})$ is a Sufficient Explanation for probability $\pi$, if $SDP_{S_\pi(\boldsymbol{x})}(y; \boldsymbol{x}, t) \ge \pi$, and no subset $Z$ of $S_\pi(\boldsymbol{x})$ satisfies $SDP_Z(y; \boldsymbol{x}, t) \ge \pi$. Hence, a Minimal Sufficient Explanation is a Sufficient Explanation with minimal size.

For a given instance, the Sufficient Explanation or Minimal Sufficient Explanation may not be unique [Darwiche and Hirth, 2020]. Furthermore, there may be significant differences among the Sufficient Explanations or Minimal Sufficient Explanations. We denote A-SE as the set of all Sufficient Explanations and M-SE as the set of Minimal Sufficient Explanations. Thus, the number and the diversity of the explanations make the method less intelligible, as deriving one of them is not informative enough, and all of them are too complex to interpret. Therefore, we propose to compute the following local attributions that summarize the importance of each variable in A-SE/M-SE:

**Definition 3.3. (Local eXplanatory Importance - LXI).** Given an instance $(\boldsymbol{x}, y)$ and its A-SE or M-SE. The local explanatory importance of $\boldsymbol{x}_i$ is how frequent $\boldsymbol{x}_i$ is chosen in the A-SE or M-SE.

Contrary to classical local feature attributions like SHAP or LIME, the values of Local Explanatory Importance does not depend on the range of values of the predictions, and are interpretable by design. It corresponds to the frequency of apparition in the A-SE or M-SE, which allows to reason about the relative difference between the attribution of each feature. Indeed, we can easily discriminate between the importance of variables in terms of probabilities compared to arbitrary values of SHAP or LIME that depend on the model and its predictions. In our framework, a value equal to 1 means that this feature is present in all the A-SE/M-SE. Hence this feature is necessary to maintain the prediction. Moreover, the attributions of the features are sparse since they are based on the A-SE/M-SE.

Although Sufficient Explanations allow finding local relevant variables, we may want to know the logical reasons relating input and output. In essence, explaining a decision means giving the reasons that highlight why the decision has been made. Therefore, we propose to extend

the Sufficient Explanations into local rules. A rule is a simple IF-THEN statement, e.g., IF the conditions on the features are met, THEN make a specific prediction. Recall that given an instance $\boldsymbol{x}$, a Sufficient Explanation is the minimal subset $S \subset [p]$, such that fixing the values $\boldsymbol{X}_S = \boldsymbol{x}_S$ permits to maintain the prediction with high probability. The idea is to find the largest rectangle $L_S(\boldsymbol{x}) = \prod_{i=1}^{|S|}[a_i, b_i], a_i, b_i \in \mathbb{R}$ given the indexes of the Sufficient Explanation $S$ such that $\boldsymbol{x}_S \in L_S(\boldsymbol{x})$ and $\forall \boldsymbol{z}$ with $\boldsymbol{z}_S \in L_S(\boldsymbol{x}), SDP_S(y; \boldsymbol{z}; t) \geq \pi$.

**Definition 3.4.** (**Minimal Sufficient Rule**). Given an instance $(\boldsymbol{x}, y)$, $S$ a Minimal Sufficient Explanation, the rectangle $L_S(\boldsymbol{x}) = \prod_{i=1}^{|S|}[a_i, b_i], a_i, b_i \in \mathbb{R}$ is a Minimal Sufficient Rule if $L_S(\boldsymbol{x}) = \operatorname{argmax}_{L(\boldsymbol{x})} Vol(L(\boldsymbol{x})), \boldsymbol{x}_S \in L_S(\boldsymbol{x})$ and $\forall \boldsymbol{z}, \boldsymbol{z}_S \in L_S(\boldsymbol{x}), SDP_S(y; \boldsymbol{z}, t) \geq \pi$.

Intuitively, the Sufficient Rule is a generalization of the Sufficient Explanation, i.e., instead of satisfying the minimality/sufficiency conditions of definition 3.2 if we fixed the values $\boldsymbol{X}_S = \boldsymbol{x}_S$, we want to satisfy these conditions on all the elements of a rectangle $L_S(\boldsymbol{x})$ that contains $\boldsymbol{x}_S$. We also want this rectangle to be of maximal volume such that it covers a large part of the input space. Thus, the Sufficient Rule captures the local behavior of the model around $\boldsymbol{x}$ while ensuring the minimality of the rule and guarantees on the outcome. Note that the volume of the rectangle $L$ can be defined as $Vol(L(\boldsymbol{x})) = P(\boldsymbol{X} \in L(\boldsymbol{x}))$ or $\lambda(L(\boldsymbol{x}))$, with $\lambda$ the Lebesgue measure.

While Sufficient Rules are similar to Anchors introduced by Ribeiro et al. [2018], we emphasize two major types of differences. The first is that our framework for constructing rules can address regression problems, deal with continuous features without discretization, and do not need access to the model $f$. Moreover, if we have a model $f$ and an instance $\boldsymbol{x}$, Anchors search the largest rule (or rectangle) $L_S(\boldsymbol{x})$ such that $P_Q(f(\boldsymbol{x}) = Y | \boldsymbol{X}_S \in L_S(\boldsymbol{x})) \geq \pi$ under an instrumental distribution $Q$. This is different from the Sufficient Rule that requires the stability of the prediction for all the observations in the rectangle i.e $\forall \boldsymbol{x}_S \in L_S(\boldsymbol{x}), P(f(\boldsymbol{x}) = Y | \boldsymbol{X}_S = \boldsymbol{x}_S) \geq \pi$. The second major difference is that the Sufficient Rule is based on the original distribution $P_{(\boldsymbol{X}, Y)}$ as we use conditional distribution $P(Y | \boldsymbol{X}_S)$. At the contrary, anchors use local sampling perturbations (introducing another distribution $Q$). As we discuss in the next section, the effective computation of these rules is very different. Anchors use a heuristic approach to find the minimal rule, which might produce suboptimal minimal rules. The Sufficient Rules satisfy a minimality principle by definition.

## 4 SDP, Sufficient Explanations and Sufficient Rules via Random Forest

In order to find the Sufficient Explanations $S_\pi(\boldsymbol{x})$ and the corresponding Sufficient Rules $L_{S_\pi}(\boldsymbol{x})$, we need to compute the SDP for any subset $S$. However, the computation of the SDP is known to be computationally hard; even for simple Naive Bayes model, the computation of SDP is NP-hard [Chen et al., 2013]. Consequently, approximate criteria based on expectations instead of probabilities have been introduced by Wang et al. [2020]. They proposed to use a Probabilistic Circuit [Choi et al., 2020] to model the distribution of the features $\boldsymbol{X}$ and compute a lower bound of the SDP.

In this section, we propose a consistent estimator of the SDP for any distribution $(\boldsymbol{X}, Y)$. It is based on two ideas: Projected Forest [Bénard et al., 2021a,c] and Quantile Regression Forest [Meinshausen and Ridgeway, 2006]. The Projected Forest is an adaptation of the Random Forest algorithm that estimates $E[Y | \boldsymbol{X}_S = \boldsymbol{x}_S]$ instead of $E[Y | \boldsymbol{X} = \boldsymbol{x}]$, and the Quantile Regression Forest uses the Random Forest algorithm to estimate the Conditional Distribution Function (CDF) $P(Y \leq y | \boldsymbol{X} = \boldsymbol{x})$. The first step is to write the SDP as

$$SDP_S(y; \boldsymbol{x}, t) = P((Y - y)^2 \leq t | \boldsymbol{X}_S = \boldsymbol{x}_S) = F_S(y + \sqrt{t} | \boldsymbol{X}_S = \boldsymbol{x}_S) - F_S(y - \sqrt{t} | \boldsymbol{X}_S = \boldsymbol{x}_S).$$

Equation 4.1 shows that the only challenge is to estimate the Projected (or Conditional) CDF $F_S(y | \boldsymbol{X}_S = \boldsymbol{x}_S) = P(Y \leq y | \boldsymbol{X}_S = \boldsymbol{x}_S)$. The variant of the original Random Forest proposed by Meinshausen and Ridgeway [2006] that estimates the CDF $F(y | \boldsymbol{X} = \boldsymbol{x}) = P(Y \leq y | \boldsymbol{X} = \boldsymbol{x})$ is not of interest to us because we want to estimate the Projected CDF $F_S(y | \boldsymbol{X}_S = \boldsymbol{x}_S)$ for any $S$. The recent works by Bénard et al. [2021a,c] are much more relevant as they permit to estimate $E[Y | \boldsymbol{X}_S = \boldsymbol{x}_S]$ from a classical Random Forest that has learned to predict $E[Y | \boldsymbol{X} = \boldsymbol{x}]$. The idea is to extract a new Forest called Projected Forest from the original Forest, which is a projection of the original Forest along the $S$-direction.

We propose to combine the ideas of Quantile Regression Forest and Projected Forest to estimate the Projected CDF $F_S(y | \boldsymbol{X}_S = \boldsymbol{x}_S)$. In addition, we prove the consistency of our estimator of the Projected CDF.

## 4.1 Random Forest and Condition Distribution Function (CDF) Forest

A Random Forest (RF) is grown as an ensemble of $k$ trees, based on random node and split point selection based on the CART algorithm [Breiman et al., 1984]. The algorithm works as follows. For each tree, $a_n$ data points are drawn at random with replacement from the original data set of size $n$; then, at each cell of every tree, a split is chosen by maximizing the CART-criterion; finally, the construction of every tree is stopped when the total number of cells in the tree reaches the value $t_n$. For each new instance $\boldsymbol{x}$, the prediction of the $l$-th tree is:

$$m_n(\boldsymbol{x}, \Theta_l, \mathcal{D}_n) = \sum_{i=1}^{n} \frac{B_n(\boldsymbol{X}^i; \Theta_l)\, \mathbb{1}_{X^i \in A_n(\boldsymbol{x};\Theta_l,\mathcal{D}_n)}}{N_n(\boldsymbol{x};\Theta_l,\mathcal{D}_n)} Y^i \qquad (4.1)$$

where:
- $\Theta_l, l = 1, \ldots, k$ are independent random vectors, distributed as a generic random vector $\Theta = (\Theta^1, \Theta^2)$ and independent of $\mathcal{D}_n$. $\Theta^1$ contains indexes of observations that are used to build the tree, i.e. the bootstrap sample and $\Theta^2$ indexes of splitting candidate variables in each node. $\Theta_{1:k}$ denotes the sequence of $\Theta_l$'s.

- $A_n(\boldsymbol{x};\Theta_l,\mathcal{D}_n)$ is the tree cell (leaf) containing $\boldsymbol{x}$

- $N_n(\boldsymbol{x};\Theta_l,\mathcal{D}_n)$ is the number of bootstrap elements that fall into $A_n(\boldsymbol{x};\Theta_l,\mathcal{D}_n)$

- $B_n(\boldsymbol{X}^i;\Theta_l)$ is the bootstrap component i.e. the number of times that the observation has been chosen from the original data.

The trees are then averaged to gives the prediction of the forest as:

$$m_{k,n}(\boldsymbol{x}, \Theta_{1:k}, \mathcal{D}_n) = \frac{1}{k} \sum_{l=1}^{k} m_n(\boldsymbol{x};\Theta_l,\mathcal{D}_n) \qquad (4.2)$$

The Random Forest estimator (Eq. 4.2) can also be seen as an adaptive neighborhood procedure [Lin and Jeon, 2006]. For every instance $\boldsymbol{x}$, the observations in $\mathcal{D}_n$ are weighted by $w_{n,i}(\boldsymbol{x};\Theta_{1:k},\mathcal{D}_n)$, $i = 1, \ldots, n$. Therefore, the prediction of Random Forests and the weights can be rewritten as

$$m_{k,n}(\boldsymbol{x}, \Theta_{1:k}, \mathcal{D}_n) = \sum_{i=1}^{n} w_{n,i}(\boldsymbol{x};\Theta_{1:k},\mathcal{D}_n)Y^i, \quad w_{n,i}(\boldsymbol{x};\Theta_{1:k},\mathcal{D}_n) = \sum_{l=1}^{k} \frac{B_n(X^i;\Theta_l)\, \mathbb{1}_{X^i \in A_n(\boldsymbol{x};\Theta_l,\mathcal{D}_n)}}{k \times N_n(\boldsymbol{x};\Theta_l,\mathcal{D}_n)}$$

Viewing a Random Forest as an adaptive nearest neighbor predictor offers natural estimates of more complex quantities (Cumulative hazard function [Ishwaran et al., 2008], Treatment effect [Wager and Athey, 2017], and conditional density [Du et al., 2021]). Therefore, just as $E[Y|\boldsymbol{X} = \boldsymbol{x}]$ is approximated by a weighted mean over observation of $Y^i$, $E[\mathbb{1}_{Y \leq y}|\boldsymbol{X} = \boldsymbol{x}]$ is approximated by the weighted mean over the observations of $\mathbb{1}_{Y^i \leq y}$ using the same weights $w_{n,i}(\boldsymbol{x};\Theta_{1:k},\mathcal{D}_n)$. The approximation is

$$\widehat{F}(y|\boldsymbol{X} = \boldsymbol{x}, \Theta_{1:k}, \mathcal{D}_n) = \sum_{i=1}^{n} w_{n,i}(\boldsymbol{x};\Theta_{1:k},\mathcal{D}_n)\mathbb{1}_{Y^i \leq y} \qquad (4.3)$$

To simplify notations, we omit $\Theta_1, \ldots, \Theta_k, \mathcal{D}_n$ and we write $\widehat{F}_S(y|\boldsymbol{X}_S = \boldsymbol{x}_S)$ for any $S$.

## 4.2 Projected Forest and Projected CDF Forest

We describe the Projected Forest (PRF) and show how it is combined with the Quantile Regression Forest to build the estimator of the Projected CDF. The PRF algorithm has been introduced in Bénard et al. [2021c,a]. The idea is to project the partition of each tree of the forest on the subspace spanned by the variables in $S$, thus we can estimate $E[Y|\boldsymbol{X}_S]$ rather than $E[Y|\boldsymbol{X}]$. The computation of these partitions for each $S$ can be computationally expensive in high dimension. However, Bénard et al. [2021a] uses a simple algorithm trick to derive efficiently the output of the Projected Forest without computing explicitly its partitions. Roughly, for computing the prediction of a tree, the algorithm ignores the splits that use variables not contained in $S$. It works as follow: when an observation is dropped down in the initial trees, and it encounters a split involving a variable $i \notin S$, the observation is sent both to the left and right children nodes. As a result, each observation falls in multiple terminal leaves of the tree. Thus, to compute the prediction of an instance $\boldsymbol{x}_S$, we collect the set of terminal leaves where it falls, and average the output $Y^i$ of the training observations which belong to every terminal leaf of this collection. $E[Y|\boldsymbol{X}_S = \boldsymbol{x}_S]$ is estimated as the average outputs of the training observations in the intersection of the leaves where $\boldsymbol{x}_S$ falls.

An efficient implementation of the PRF algorithm is detailed in Appendix. The associated PRF is $m_{k,n}^{(\boldsymbol{x}_S)}(\boldsymbol{x}_S) = \sum_{i=1}^{n} w_{n,i}(\boldsymbol{x}_S) Y^i$ where the weights are defined by

$$w_{n,i}(\boldsymbol{x}_S) = \sum_{l=1}^{k} \frac{B_n(X^i;\Theta_l)\, \mathbb{1}_{X^i \in A_n^{(\boldsymbol{x}_S)}(\boldsymbol{x}_S;\Theta_l,\mathcal{D}_n)}}{k \times N_n^{(\boldsymbol{x}_S)}(\boldsymbol{x};\Theta_l,\mathcal{D}_n)}, \tag{4.4}$$

where $A_n^{(\boldsymbol{x}_S)}(\boldsymbol{x}_S;\Theta_l,\mathcal{D}_n)$ is the leaf of the associated Projected $l$-th tree where $\boldsymbol{x}_S$ falls and $N_n^{(\boldsymbol{x}_S)}(\boldsymbol{x};\Theta_l,\mathcal{D}_n)$ is the number of bootstrap observations that falls in $A_n^{(\boldsymbol{x}_S)}(\boldsymbol{x}_S;\Theta_l,\mathcal{D}_n)$. Therefore, we approximate the Projected CDF $F_S(y|\boldsymbol{X}_S = \boldsymbol{x}_S) = P(Y \leq y|\boldsymbol{X}_S = \boldsymbol{x}_S)$ as in Eq. 4.3 by using the weights of the Projected Forest defined in Eq. 4.4. The estimator of the Projected CDF is defined as: $\widehat{F}_S(y|\boldsymbol{X}_S = \boldsymbol{x}_S) = \sum_{i=1}^{n} w_{n,i}(\boldsymbol{x}_S) \mathbb{1}_{Y^i \leq y}$.

## 4.3 Consistency of the Projected CDF Forest

In this section, we state our main result, which is the uniform a.s. convergence of the estimator $\widehat{F}_S(y|\boldsymbol{X}_S = \boldsymbol{x}_S)$ to $F_S(y|\boldsymbol{X}_S = \boldsymbol{x}_S)$. Meinshausen and Ridgeway [2006] showed the uniform convergence in probability of a simplified version of the estimator of the CDF defined in Eq. 4.3, where the weights $w_{n,i}(\boldsymbol{x}_S;\Theta_{1:k},\mathcal{D}_n)$ are in fact considered to be non-random while they are indeed random variables depending on $(\Theta_l)_{l=1,\dots,k}, \mathcal{D}_n$. Moreover, the bootstrap was replaced by subsampling without replacement as in most studies that analyse the asymptotic properties of Random Forests [Scornet et al., 2015, Wager and Athey, 2017, Goehry, 2020]. However, Elie-Dit-Cosaque and Maume-Deschamps [2020] showed the almost everywhere uniform convergence of both estimators (the simplified and the one defined in Eq. 4.3) under realistic assumptions with all the randomness and bootstrap samples. We follow their works to prove the consistency of the PRF CDF $\widehat{F}_S(y|\boldsymbol{X}_S = \boldsymbol{x}_S)$ based on the following assumptions.

**Assumption 4.1.** $\forall x \in \mathbb{R}^d$, the conditional cumulative distribution function $F(y|X = x)$ is continuous.

Assumption 4.1 is necessary to get uniform convergence of the estimator.

**Assumption 4.2.** For $l \in [k]$, we assume that the variation of the conditional cumulative distribution function within any cell goes to 0.

$$\forall x \in \mathbb{R}^d, \forall y \in \mathbb{R}, \quad \sup_{\boldsymbol{z} \in A_n(\boldsymbol{x};\Theta_l,\mathcal{D}_n)} |F(y|\boldsymbol{z}) - F(y|\boldsymbol{x})| \overset{a.s}{\to} 0$$

Assumption 4.2 allows to control the approximation error of the estimator. If for all $y$, $F(y|.)$ is continuous, Assumption 4.2 is satisfied provided that the diameter of the cell goes to zero. Note that the vanishing of the diameter of the cell is a necessary condition to prove the consistency of general partitioning estimator (see chapter 4 in Györfi et al. [2002]). Scornet et al. [2015] show that it is true in RF where the bootstrap step is replaced by subsampling without replacement and the data come from additive regression models [Stone, 1985]. The result is also valid for all regression functions, with a slightly modified version of RF, where there are at least a fraction $\gamma$ observations in children nodes, and the number of splitting candidate variables is set to 1 at each node with a small probability. Under these small modifications, Lemma 2 from Meinshausen and Ridgeway [2006] gives that the diameter of each cell vanishes.

**Assumption 4.3.** Let $k$ and $N_n(\boldsymbol{x};\Theta_l,\mathcal{D}_n)$ (number of bootstrap observations in a leaf node), then there exists $k = \mathcal{O}(n^\alpha)$, with $\alpha > 0$, and $\forall \boldsymbol{x} \in \mathbb{R}^d, N_n(\boldsymbol{x};\Theta_l,\mathcal{D}_n) = \Omega^2(\sqrt{n}(ln(n))^\beta)$, with $\beta > 1$ a.s.

Assumption 4.3 allows us to control the estimation error and means that the cells should contain a sufficiently large number of points so that averaging among the observations is effective.

To prove the consistency of the PRF CDF $\widehat{F}_S(y|\boldsymbol{X}_S = \boldsymbol{x}_S)$, we only need to verify the assumptions 4.1, 4.2, 4.3 on the parameters of the PRF CDF and the Projected CDF $F_S(y|\boldsymbol{X}_S = \boldsymbol{x}_S) = P(Y \leq y|\boldsymbol{X}_S = \boldsymbol{x}_S)$.

Assumptions 4.1 and 4.2 are satisfied for the Projected CDF and the PRF CDF's leaves. Since by definition $A_n^{(\boldsymbol{x}_S)}(\boldsymbol{x}_S;\Theta_l,\mathcal{D}_n) \subset A_n(\boldsymbol{x};\Theta_l,\mathcal{D}_n)$, if the variations within the cells of the RF

---

$^2 f(n) = \Omega(g(n)) \iff \exists k > 0, \exists n_0 > 0|\ \forall n \geq n_0 |f(n)| \geq |g(n)|$

vanish (diameter goes to zero), it also vanishes in the Projected forest. In addition, if the CDF $F(y|\boldsymbol{X} = \boldsymbol{x}) = F(y|\boldsymbol{X}_S = \boldsymbol{x}_S, \boldsymbol{X}_{\bar{S}} = \boldsymbol{x}_{\bar{S}})$ is continuous, we can show by a straightforward analysis of parameter-dependent integral that the Projected CDF $F_S(y|\boldsymbol{X}_S = \boldsymbol{x}_S) = \int F(y|\boldsymbol{X}_S = \boldsymbol{x}_S, \boldsymbol{X}_{\bar{S}} = \boldsymbol{x}_{\bar{S}})p(\boldsymbol{x}_{\bar{S}}|\boldsymbol{x}_S)d\boldsymbol{x}_{\bar{S}}$ is also continuous. As we control the minimal number of observations in the leaf of the PRF CDF by construction, Assumption 4.3 is also verified. Then, the PRF CDF satisfies also Assumption 4.1-4.3 which ensures its consistency thanks to Theorem 4.4.

**Theorem 4.4.** *Consider a RF satisfying Assumptions 4.1 to 4.3. Then,*

$$\forall \boldsymbol{x} \in \mathbb{R}^d, \sup_{y \in \mathbb{R}} |\widehat{F}_S(y|\boldsymbol{X}_S = \boldsymbol{x}_S) - F_S(y|\boldsymbol{X}_S = \boldsymbol{x}_S)| \overset{a.s}{\to} 0$$

The complete proof and simulations showing the convergence of the estimator can be found in Appendix.

### 4.4 Estimation of SDP, Sufficient Explanations and Sufficient Rules

In this section, we show how we compute the SDP, Sufficient Explanations, and Sufficient Rules using the PRF CDF estimator. We derive from the previous section the following consistent estimator of any $SDP_S(y; \boldsymbol{x}, t)$:

$$\widehat{SDP}_S = \widehat{F}_S(y + \sqrt{t}\,|\boldsymbol{X}_S = \boldsymbol{x}_S) - \widehat{F}_S(y - \sqrt{t}\,|\boldsymbol{X}_S = \boldsymbol{x}_S)$$

However, finding the A-SE/M-SE using a greedy algorithm is computationally hard, since the number of subsets is exponential. Therefore, we propose to reduce the number of variables by focusing only on the most influential variables. We search the Sufficient Explanations in the subspace of the $s = 10$ variables frequently selected in the RF used to estimate the SDP, reducing the complexity from $2^p$ to $2^{10}$. This preselection procedure is already used in Bénard et al. [2021b,a], and it is mainly based on Proposition 1 of Scornet et al. [2015], which highlights the fact that RF naturally splits the most on influential variables. Note that the minimum number of selected variables $s$ is a hyperparameter.

To find the Sufficient Rules, we used the SDP's estimator $\widehat{SDP}_S$. By using the fact that the PRF CDF is a tree-based model, and thus $\widehat{SDP}_S$ partitions the space like a tree or a Random Forest, we do not need to discretize the continuous space to find the largest rectangle. We only need to find the leaves compatible with the conditions of the Sufficient Rule defined in 3.4. Given a Minimal Sufficient Explanation $S$ of an instance $\boldsymbol{x}$, we already a have a rectangle $L_S(\boldsymbol{x})$ defined by the PRF CDF or $\widehat{SDP}_S$ that is the largest rectangle such that $\boldsymbol{x}_S \in L_S(\boldsymbol{x})$ and $\forall \boldsymbol{z}$ with $\boldsymbol{z}_S \in L_S(\boldsymbol{x})$, $\widehat{SDP}_S(y; \boldsymbol{z}, t) = \widehat{SDP}_S(y; \boldsymbol{x}, t) \geq \pi$. By definition, it is the intersection of the cell of the trees where $\boldsymbol{x}_S$ falls, namely $\cap_{l=1}^k A_n^{(\boldsymbol{x}_S)}(\boldsymbol{x}_S; \Theta_l, \mathcal{D}_n)$. Thus, starting from $\cap_{l=1}^k A_n^{(\boldsymbol{x}_S)}(\boldsymbol{x}_S; \Theta_l, \mathcal{D}_n)$, which is also a cell (leaf) of the Projected Forest, we can find all the neighboring leaf (rectangle) that we can merge with it to get the largest rectangle. We will see in the next section that it provides good insights about the local behaviour of the model.

**How to choose the hyperparameters.** The main hyperparameters are: $s$ the number of preselected variables, $\pi$ the minimal probability of changing the decision and $t$ which corresponds to the radius of the ball center at the prediction in the definition 3.1 of the SDP for regression problems.

The choice of $s \ll p$ is motivated by the fact that many datasets have intrinsic dimensions much lower than the ambient dimension. Our Selection criterion is based on Proposition 1 in [Scornet et al., 2015], which highlights the fact that RF naturally splits the most on influential variables. However, any RF's importance measure s.t. Mean Decrease Accuracy [Breiman, 2001] or Impurity [Breiman and Cutler, 2003] can be used as the RF algorithm is known to adapt to the intrinsic dimension [Scornet et al., 2015, Klusowski, 2020]. Therefore, the choice of $s$ is directly driven by the computation power available to explore the subsets. In practice, we have always found coalitions $S$ with a probability above $\pi = 0.9$ with $s = 10$ for real world datasets. Otherwise, we suggest a trade-off between the maximal reachable size $s$ and the highest probability.

We propose choosing $\pi = 0.9$ as it is an acceptable level of risk, but the user can increase/decrease this probability depending on the use case.

The most challenging hyperparameter to select is $t$; we recommend having an adaptive radius $t(x)$ using the quantile of the conditional distributions $Y|X = x$, which is a by-product of the

Quantile Regression Forest used for computing the SDP. For each observation, we choose the radius $t(\boldsymbol{x}) = [\hat{q}_{\alpha_1}(\boldsymbol{x}), \hat{q}_{1-\alpha_2}(\boldsymbol{x})]$ with $\alpha_1 + \alpha_2 = \alpha$. We build then a confidence interval of varying length but with constant confidence level across the dataset. This makes our approach as a natural generalization of the SDP in the classification case by accounting for the uncertainty of the model to explain. In that case, the SDP and associated sufficient coalition should be read: "if $\boldsymbol{X}_S = \boldsymbol{x}_S$ is fixed, then there is a probability at least $\pi$ of not changing the prediction significantly with level $1 - \alpha$, or so that $Y \in [\hat{q}_{\alpha_1}(\boldsymbol{x}), \hat{q}_{1-\alpha_2}(\boldsymbol{x})]$ with probability $\pi$. For this reason, we suggest fixing $\alpha_1 + \alpha_2 = \alpha$ and $\pi$ at standard level $1 - \alpha = \pi = 0.9$ agreeing with acceptable level of risks.

## 5 Experiments

We conduct three experiments in this section. The first compares the Sufficient Explanations, Sufficient Rules and LXI with state-of-art (SOTA) local explanations methods (SHAP, LIME, INVASE) in a simple high dimensional regression model with small relevant features. Although this model is simple, SOTA (SHAP, LIME) have been shown to poorly detect the important variables of this model [Amoukou et al., 2021, Ghalebikesabi et al., 2021]. Then, we analyse the performance of the Sufficient Rules in a real world regression problem. Finally, we highlight the advantages of the Sufficient Rules in comparison with Anchors in real world classification datasets. More experiments can be found in Appendix.

To effectively compare different explanation methods, we use synthetic data since we need the ground truth. We use the following synthetic model: we have $\boldsymbol{X} \in \mathbb{R}^p$, $\boldsymbol{X} \in \mathcal{N}(0, \Sigma)$, $\Sigma = 0.8J_p + 5I_p$ with $p = 100$, $I_p$ is the identity matrix, $J_p$ is all-ones matrix and a linear predictor with switch defined as:

$$Y = (X_1 + X_2)\mathbb{1}_{X_5 \leq 0} + (X_3 + X_4)\mathbb{1}_{X_5 > 0}. \tag{5.1}$$

The variables $X_i$ for $i = 6 \ldots 100$ are noise variables. We fit a RF with a sample size $n = 10^4$, $k = 20$ trees and the minimal number of samples by leaf node is set to $t_n = \lfloor \sqrt{n} \times \ln(n)^{1.5}/250 \rfloor$ for the original and the Projected Forest. The $R^2 = 99\%$ on the test set of size $10^4$. The RF is used to compute the explanations of SHAP, LIME. The Projected Forest is also extract from the RF for the SDP approaches. We choose $\alpha_1 = 0.05, \alpha_2 = 0.95$ and $\pi = 0.90$. For INVASE, we use Neural Networks with 3 hidden layers for the selector model and the predictor model as in Yoon et al. [2018]. Notice that for SHAP, LIME and the SDP approaches, we used the same information (the learned RF) to retrieve the true explanation of the data. The performance of INVASE and the RF is the same, both model perfectly fit the data with a $R^2 = 99\%$.

**SDP approaches vs SOTA (SHAP, LIME, INVASE) on regression.**     Here, we analyze the capacity of each method to discover the local important variables of the model defined in Eq. 5.1. Indeed, Eq. 5.1 shows that if $x_5 \leq 0$, the model uses only the variables $x_1, x_2$ otherwise it uses the variables $x_3, x_4$. Thus, we try to find the top $K = 3$ relevant features for each sample. Note that $K$ is not a required input for SDP and INVASE, but $K$ must be given for SHAP and LIME. We use the true positive rate (TPR) (higher is better) and false discovery rate (FDR) (lower is better) to measure the performance of the methods on discovery (i.e., discovering which features are relevant). In addition, as one of the objectives of each method is to find the minimal subset $\boldsymbol{x}_S$ that is relevant to the corresponding target $y$, we also compute a predictive performance metrics that shows how well the projected predictor $E[Y|\boldsymbol{X}_S = \boldsymbol{x}_S]$ selected by each method is close to the predictor on the full set of features $E[Y|\boldsymbol{X} = \boldsymbol{x}]$, under the data distribution. Formally, for a given subset $S$, we denote it as P-MSE $= E_Z \left[ \left( E[Y|\boldsymbol{X} = \boldsymbol{Z}] - E[Y|\boldsymbol{X}_S = \boldsymbol{Z}_S] \right)^2 \right]$ where $\boldsymbol{Z} \sim P_{\boldsymbol{X}}$.

We obtain the following results for **Sufficient Explanation** (TPR= 99%, FDR= 2%, P-MSE= 0.02), **INVASE** (TPR= 99%, FDR= 87%, P-MSE= 0.006 ), **SHAP** (TPR= 73%, FDR= 27%, P-MSE= 0.79), and **LIME** (TPR= 50%, FDR= 49%, P-MSE= 5.01). We observe that the Sufficient Explanation succeeds to find the top $K$ relevant variables and outperform the other methods by a significant margin. SHAP and LIME obtain the worst discovery rate. INVASE succeeds in finding the relevant variables, but it has a high FDR (87%), which means we cannot distinguish between the relevant and irrelevant variables since 87% of the selected variables are irrelevant. We also see that the P-MSE of INVASE is the lowest, which is not surprising as it selects all the relevant variables despite its high FDR. Indeed, this metric is not much affected by the FDR. The P-MSE of Sufficient

Explanations is also almost zero, and as above, SHAP and LIME perform worse than the other methods. In table 1, we compare the LXI and SHAP values on 1000 observations having $X_5 > 0$. We compare the mean absolute values of SHAP and average LXI on this sub-population. Notice that on this model, these observations have a single Sufficient Explanation which is the variables $X_3, X_4, X_5$. Both models give null attributions to the noise variables, but SHAP gives higher importance to the variables $X_1, X_2$ than the truly important variables $X_3, X_4, X_5$. On the other hand, LXI gives non null attributions only on the important variables. We refer to the Appendix for an additional comparison with SHAP in a case where there are several Sufficient Explanations.

Table 1: Global SHAP values (mean absolute) and average LXI on 1000 observations of the test set having $X_5 > 0$. $X_{noises}$ corresponds to the sum of the attributions of the noises variables ($X_i$ for $i = 6 \ldots 100$).

|      | $X_1$ | $X_2$ | $X_3$ | $X_4$ | $X_5$ | $X_{noises}$ |
|------|-------|-------|-------|-------|-------|--------------|
| LXI  | 0     | 0     | 1     | 1     | 1     | 0            |
| SHAP | 1.47  | 1.54  | 0.56  | 0.56  | 0.86  | 0.005        |

However, even if the Sufficient Explanation find effectively the top $K$ relevant variables, it cannot provide a complete understanding of the local behavior of the regression model (the SOTA methods also), i.e., that it's the sign of $x_5$ that matters. Thus, by extending the Sufficient Explanation into Sufficient Rule we can retrieve the complete story. We choose an observation $(\boldsymbol{x}, y)$ such that its Sufficient Explanation found is $S = [X_3, X_4, X_5]$, with $\boldsymbol{x}_S = [-3.64, -4.41, 0.68]$. Although the Sufficient Explanation shows that fixing the value $\boldsymbol{x}_S$ permit to maintain the prediction with high probability, the Sufficient Rule gives the additional information that we can also maintain the prediction within a small radius around $y$ by satisfying the rule $L_S(\boldsymbol{x}) = \{X_5 > 0 \text{ AND } -4.45 \leq X_4 \leq -4.06 \text{ AND } -3.67 \leq X_3 \leq -3.58\}$. The Sufficient Rule $L_S(\boldsymbol{x})$ catches perfectly the local behaviour of the model which says that despite the values of $x_3, x_4$, it's the sign of $x_5$ that matters.

**SDP approaches on real world regression.** We demonstrate the performance and flexibility of the Sufficient Rules (SR) on a real world regression dataset. Since there are no ground truth explanations for real world datasets, we use the predictive performance and simplicity (number of variables used) of the SR as an indicator of the effectiveness of the explanations. Indeed, we can build a global model by combining all the Sufficient Rules found for the observations in the training set, and we measure its performance on the test set. We set the output of each rule as the majority class (resp. average values) for classification (resp. regression) of the training observations that satisfy this rule. Note that some rules can overlap and an observation can satisfy multiple rules. To resolve these conflicts, we use the output of the rule with the best precision (AUC or MSE). We called this model Global-SR. We have experimented on Bike Sharing data [Kaggle, 2015] that contains 10886 records and 15 variables about historical usage patterns with weather data in order to forecast bike rental demand in the Capital Bikeshare program in Washington, D.C.

We split the data into train (75%) - test (25%) set and train a RF with $k = 20$ trees and maximal depth = 14. It has mean absolute error $MAE = 25$ and $R^2 = 94\%$ on test set. We use the RF on the train set to generate the SR. Although the Global-SR covers 78% of the test set, we observe that it performs as well as the baseline model with $MAE = 29$, $R^2 = 90\%$ while being transparent in its decision process. Note that the rules of the SR on Bike Sharing Demand are based on 4.5 variables in average. We give examples of the learned rules: $R_1 = \{$If Workingday = True and Hours $\in$ [5.5, 6.5] THEN Bike rental demand = 20$\}$, $R_2 = \{$If Hours $\in$ [8.5, 9] and Year $\leq$ 2011 and month $\geq$ 5 THEN Bike rental demand = 192$\}$. The number of observations that satisfy these rules is 134, 133 for the rules $R_1$, $R_2$ respectively with a mean absolute error $MAE_{R_1} = 12, MAE_{R_2} = 30$.

**Anchors vs Sufficient Rules (SR).** To compare our methods w.r.t to Anchors, we have to consider a classification problem. We use three popular real world datasets: **Compas** ($n = 6167, p = 14$)[Larson et al., 2016], **Nhanesi** ($n = 8593, p = 17$) [CDC, 1999-2022], and **Employee Attrition** ($n = 1470, p = 27$) [Kaggle, 2017] which we split into train (75%) - test (25%) set, and train a RF with the parameters of the previous section. We use the RF to generate the local rule-based explanations with Anchors and the SDP approach (Sufficient Rules) to explain the RF's predictions on the test set. We aim to evaluate the generalization of each explanation across the population. Thus, we measure the following metrics, *Coverage* (higher is better): what fraction of unseen instances fall

in the rule and *Correctness* (higher is better): average number of unseen instances that satisfy the rule and has the same output than the observation that generate the rule, *Sparsity* (lower is better): the mean, variance and maximal size of the rule (number of variable on which it is based).

Table 2: Results of the *Correctness* (Acc), *Coverage* (Cov), and *Sparsity* (Sprs) on **Compas**, **Nhanesi**, **ATTRITION** of the Sufficient Rules (SR) and Anchors. The vector of Sprs (mean, std, max) corresponds to the mean, variance, and the max values of the rules.

| | **COMPAS** | | | **NHANESI** | | | **ATTRITION** | | |
|---|---|---|---|---|---|---|---|---|---|
| | Acc | Sprs | Cov | Acc | Sprs | Cov | Acc | Sprs | Cov |
| SR | 0.95 | (1.6, 0.96, 7) | 0.30 | 0.97 | (1.3, 0.65, 7) | 0.41 | 0.95 | (1.15, 0.90, 9) | 0.76 |
| Anchors | 0.92 | (1.83, 1.89, 11) | 0.23 | 0.96 | (1.8, 3.91, 16) | 0.31 | 0.95 | (0.82, 4.24, 21) | 0.74 |

In table 2, we observe that both model have a high accuracy in all datasets, but SR consistently outperforms Anchors on all datasets.

On the other hand, Anchors uses many more features. Indeed, by sampling marginally (i.e. assuming that the features are independent) Anchors can succeed to find accurate and high coverage rule, but at the cost of optimality. In fact, we can observe in table 2 that Anchors tends to give much longer rules. While the observed maximal size of SR is 9 in all dataset, Anchors can provide a rule of size 12 (**Compas**), 16 (**Nhanesi**), 23 (**Attrition**). For instance, the size distribution of Anchors on **Nhanesi** is represented with the following dictionary {**size** : $count$}: {**1** : 704, **2** : 127, **3** : 71, **4** : 21, **5** : 13, **6** : 10, **7** : 10, **8** : 9, **9** : 9, **10** : 4, **11** : 2, **12** : 4, **13** : 5, **14** : 1, **16** : 1}, and the corresponding distribution for the SR is {**1** : 775, **2** : 145, **3** : 52, **4** : 9, **5** : 12, **7** : 4}. Note that this is a significant drawback of Anchors, as simplicity of the explanations is an essential desideratum for explanation methods. We give an additional experiment confirming these results in the Appendix.

Another desirable property of explanation methods is stability, i.e., nearby observations must have the same explanations. Here, we evaluate the stability of the methods w.r.t input perturbations. For each observation $x$, we compare its rule with the rules of 50 noisy versions of $x$ obtained by adding random Gaussian noises $\mathcal{N}(0, \epsilon \times I)$ to the values of the features with $\epsilon = 0.1$. The perturbation is small enough to not change the prediction. For each dataset (**Compas, Nhanesi, Attrition**), we randomly perturb 100 observations in the test set (50 times), and we observe in average 10 (std=76), 6.83 (std=139), 14 (std=58) different rules for Anchors respectively, while we have 1.5 (std=0.25), 1.1 (std=1.9), 1.13 (std=0.13) for SR, resp. It shows the large instability of Anchor compared to SR. Indeed, even when $\epsilon = 0$, Anchors gives different rules, e.g., on **Compas** its has 7 (std=70) different rules in average with no perturbations. Results for other values of $\epsilon$ can be found in Appendix.

These experiments demonstrate that the SR gives more optimal rules than the one returned by Anchors. We also conduct an additional experiment in the Appendix confirming this claim in a setting where we know the ground truth.

## 6 Conclusion

In this work, we introduce a fast and consistent estimator of the Same Decision Probability and propose a natural generalization of the SDP for regression problems. Thus, we introduce the first local rule-based explanations for regression. We give consistent estimates of three local explanation methods: Minimal Sufficient Explanations, Local eXplanatory Importance, and Minimal Sufficient Rules for any data. We prove that these methods considerably improve local variable detection over state-of-the-art algorithms while ensuring minimality, sufficiency, and stability. Our generalization of SDP and Minimal Sufficient Rules are tightly related. They are linked by a Random Forest, which is a computationally and statistically efficient estimator of the SDP and gives the partition that is translated into an interpretable rule. Therefore, our method is principally suitable for datasets where tree-based models work (e.g., tabular data). In future works, we aim at improving the interpretability and confidence of the Sufficient Rules by taking into account uncertainty estimates of their predictions.

**Acknowledgments.** This work has been done in collaboration between Stellantis and Laboratoire de Mathématiques et Modélisation d'Évry (LaMME) and was supported by the program Convention Industrielle de Formation par la Recherche (CIFRE) of the Association Nationale de la Recherche et de la Technologie (ANRT).

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
