# Supplementary materials: Consistent Sufficient Explanations and Minimal Local Rules for explaining any classifier or regressor

**Salim I. Amoukou**
LaMME
University Paris Saclay
Stellantis Paris

**Nicolas J-B. Brunel**
LaMME
ENSIIE, University Paris Saclay
Quantmetry Paris

## Contents

36th Conference on Neural Information Processing Systems (NeurIPS 2022).

# A Proofs

In this section, we prove our main result, Theorem 4.4, which is the uniform a.s. consistency of the PRF CDF $\widehat{F}_S(y|\boldsymbol{X}_S = \boldsymbol{x}_S)$ to the projected CDF $F_S(y|\boldsymbol{X}_S = \boldsymbol{x}_S)$.

## A.1 Main assumptions

**Assumption A.1.** $\forall x \in \mathbb{R}^d$, the conditional cumulative distribution function $F(y|X = x)$ is continuous.

Assumption A.1 is necessary to get uniform convergence of the estimator.

**Assumption A.2.** For $l \in [k]$, we assume that the variation of the conditional cumulative distribution function within any cell goes to $0$.

$$\forall x \in \mathbb{R}^d, \forall y \in \mathbb{R}, \quad \sup_{\boldsymbol{z} \in A_n(\boldsymbol{x};\Theta_l,\mathcal{D}_n)} |F(y|\boldsymbol{z}) - F(y|\boldsymbol{x})| \overset{a.s}{\to} 0$$

Assumption A.2 allows to control the approximation error of the estimator. If for all $y$, $F(y|.)$ is continuous, Assumption A.2 is satisfied provided that the diameter of the cell goes to zero. Note that the vanishing of the diameter of the cell is a necessary condition to prove the consistency of general partitioning estimator (see chapter 4 in Györfi et al. [2002]). Scornet et al. [2015] show that it is true in RF where the bootstrap step is replaced by subsampling without replacement and the data come from additive regression models [Stone, 1985]. The result is also valid for all regression functions, with a slightly modified version of RF, where there are at least a fraction $\gamma$ observations in children nodes, and the number of splitting candidate variables is set to 1 at each node with a small probability. Under these small modifications, Lemma 2 from Meinshausen and Ridgeway [2006] gives that the diameter of each cell vanishes.

**Assumption A.3.** Let $k$ and $N_n(\boldsymbol{x};\Theta_l,\mathcal{D}_n)$ (number of boostrap observations in a leaf node), then there exists $k = \mathcal{O}(n^\alpha)$, with $\alpha > 0$, and $\forall \boldsymbol{x} \in \mathbb{R}^d$, $N_n(\boldsymbol{x};\Theta_l,\mathcal{D}_n) = \Omega^1(\sqrt{n}(ln(n))^\beta)$, with $\beta > 1$ a.s.

Assumption A.3 allows us to control the estimation error and means that the cells should contain a sufficiently large number of points so that averaging among the observations is effective.

To prove the consistency of the PRF CDF $\widehat{F}_S(y|\boldsymbol{X}_S = \boldsymbol{x}_S)$, we only need to verify the assumptions A.1, A.2, A.3 on the parameters of the PRF CDF and the Projected CDF $F_S(y|\boldsymbol{X}_S = \boldsymbol{x}_S) = P(Y \leq y|\boldsymbol{X}_S = \boldsymbol{x}_S)$.

Assumptions A.1 and A.2 are satisfied for the Projected CDF and the PRF CDF's leaves. Since by definition $A_n^{(\boldsymbol{x}_S)}(\boldsymbol{x}_S;\Theta_l,\mathcal{D}_n) \subset A_n(\boldsymbol{x};\Theta_l,\mathcal{D}_n)$, if the variations within the cells of the RF vanish, it also vanishes in the projected forest. In addition, if the CDF $F(y|\boldsymbol{X} = \boldsymbol{x}) = F(y|\boldsymbol{X}_S = \boldsymbol{x}_S, \boldsymbol{X}_{\bar{S}} = \boldsymbol{x}_{\bar{S}})$ is continuous, we can show by a straightforward analysis of parameter-dependent integral that the Projected CDF $F_S(y|\boldsymbol{X}_S = \boldsymbol{x}_S) = \int F(y|\boldsymbol{X}_S = \boldsymbol{x}_S, \boldsymbol{X}_{\bar{S}} = \boldsymbol{x}_{\bar{S}})p(\boldsymbol{x}_{\bar{S}}|\boldsymbol{x}_S)d\boldsymbol{x}_{\bar{S}}$ is also continuous. Since we control the minimal number of observations in the leaf of the Projected Forest by construction, Assumption A.3 is also verified. Then, the PRF CDF satisfies also Assumption A.1-A.3 which ensures its consistency thanks to Theorem 4.4.

## A.2 Proof of theorem 4.4

**Theorem A.4.** *Consider a random forest which satisfies Assumtions A.1 to A.3. Then,*

$$\forall \boldsymbol{x} \in \mathbb{R}^d, \quad \sup_{y \in \mathbb{R}} |\widehat{F}_S(y|\boldsymbol{X}_S = \boldsymbol{x}_S, \Theta_1, \ldots, \Theta_k, \mathcal{D}_n) - F_S(y|\boldsymbol{X}_S = \boldsymbol{x}_S)| \overset{a.s}{\to} 0 \qquad \text{(A.1)}$$

The main idea is to use a second independent sample $\mathcal{D}_n^\diamond$. Let assume we have a honest forest [Wager and Athey, 2017] of the Projected CDF Forest $\widehat{F}_S(y|\boldsymbol{X}_S = \boldsymbol{x}_S, \Theta_1, \ldots, \Theta_k, \mathcal{D}_n)$, which is a random

---

[1] $f(n) = \Omega(g(n)) \iff \exists k > 0, \exists n_0 > 0 | \forall n \geq n_0 |f(n)| \geq |g(n)|$

forest that grows using $\mathcal{D}_n$ but use another sample $\mathcal{D}_n^\diamond$ (independent of $\mathcal{D}_n$ and $\Theta$) to estimate the weights and the prediction. The projected CDF honest Forest is defined as:

$$F_S^\diamond(y|\boldsymbol{X}_S = \boldsymbol{x}_S, \Theta_1, \dots, \Theta_k, \mathcal{D}_n, \mathcal{D}_n^\diamond) = \sum_{i=1}^n w_{n,i}^\diamond(\boldsymbol{x}_S; \Theta_1, \dots, \Theta_k, \mathcal{D}_n, \mathcal{D}_n^\diamond) \mathbb{1}_{Y^{\diamond i} \leq y}$$

where

$$w_{n,i}^\diamond(\boldsymbol{x}_S; \Theta_1, \dots, \Theta_k, \mathcal{D}_n, \mathcal{D}_n^\diamond) = \frac{1}{k} \sum_{l=1}^k \frac{\mathbb{1}_{X^{\diamond i} \in A_n^{(S)}(\boldsymbol{x}_S; \Theta_l, \mathcal{D}_n)}}{N_n^{(S)}(\boldsymbol{x}_S; \Theta_l, \mathcal{D}_n, \mathcal{D}_n^\diamond)},$$

and $N^\diamond(A_n^{(S)}(\boldsymbol{x}_S; \Theta_l)) = N_n^{(S)}(\boldsymbol{x}; \Theta_l, \mathcal{D}_n, \mathcal{D}_n^\diamond)$ is the number of observation of $\mathcal{D}_n^\diamond = \{(X_1^\diamond, Y_1^\diamond) \dots, (X_n^\diamond, Y_n^\diamond)\}$ that fall in $A_n^{(S)}(\boldsymbol{x}_S; \Theta_l, \mathcal{D}_n)$. To ease the notations, we do not write $\Theta_1, \dots, \Theta_k, \mathcal{D}_n, \mathcal{D}_n^\diamond$ if not necessary e.g. we write $F_S^\diamond(y|\boldsymbol{X}_S = \boldsymbol{x}_S)$ instead of $F_S^\diamond(y|\boldsymbol{X}_S = \boldsymbol{x}_S, \Theta_1, \dots, \Theta_k, \mathcal{D}_n, \mathcal{D}_n^\diamond)$.

Therefore, we have $\forall \boldsymbol{x} \in \mathbb{R}^d, \forall y \in \mathbb{R}$,

$$|\widehat{F}_S(y|\boldsymbol{X}_S = \boldsymbol{x}_S) - F_S(y|\boldsymbol{X}_S = \boldsymbol{x}_S)| \leq |\widehat{F}_S(y|\boldsymbol{X}_S = \boldsymbol{x}_S) - F_S^\diamond(y|\boldsymbol{X}_S = \boldsymbol{x}_S)| + |F_S^\diamond(y|\boldsymbol{X}_S = \boldsymbol{x}_S) - F_S(y|\boldsymbol{X}_S = \boldsymbol{x}_S)|$$

The convergence of the two right-hand terms is handled separately into the following Proposition A.5 and Lemma A.1.

**Proposition A.5.** *Consider a random forest which satisfies Assumptions A.1 to A.3. Then,*

$$\forall \boldsymbol{x} \in \mathbb{R}^d, \forall y \in \mathbb{R}, \quad F_S^\diamond(y|\boldsymbol{X}_S = \boldsymbol{x}_S, \Theta_1, \dots, \Theta_k, \mathcal{D}_n, \mathcal{D}_n^\diamond) \xrightarrow[n \to +\infty]{a.s} F_S(y|\boldsymbol{X}_S = \boldsymbol{x}_S) \quad \text{(A.2)}$$

Proposition A.5 shows that the Projected CDF honest Forest is consistent and Lemma A.1 shows that the honest and the non-honest forest are close.

**Lemma A.1.** *Consider a random forest which satisfies Assumtions A.1 to A.3. Then,*

$$\forall \boldsymbol{x} \in \mathbb{R}^d, \forall y \in \mathbb{R}, \quad |F_S^\diamond(y|\boldsymbol{X}_S = \boldsymbol{x}_S, \Theta_1, \dots, \Theta_k, \mathcal{D}_n) - \widehat{F}_S(y|\boldsymbol{X}_S = \boldsymbol{x}_S, \Theta_1, \dots, \Theta_k, \mathcal{D}_n| \xrightarrow{a.s} 0$$
$$\text{(A.3)}$$

Hence, according to Proposition A.5 and Lemma A.1, we get

$$\forall \boldsymbol{x} \in \mathbb{R}^d, \forall y \in \mathbb{R}, \quad \widehat{F}_S(y|\boldsymbol{X}_S = \boldsymbol{x}_S, \Theta_1, \dots, \Theta_k, \mathcal{D}_n) \xrightarrow[n \to +\infty]{a.s} F_S(y|\boldsymbol{X}_S = \boldsymbol{x}_S) \quad \text{(A.4)}$$

To have the almost sure uniform convergence relative to y of the Projected CDF honest forest, we use Dini's second theorem. Indeed, $\{Y^{\diamond i} \leq y\} = \{U_i \leq F_S(y|X_S^{\diamond i})\}$, where $U_i, i = 1, \dots, n$ are i.i.d uniform random variables. Let $s_i = F_S(y|\boldsymbol{X}_S = X_S^{\diamond i}) = F_S(y|X_S^{\diamond i})$ and $s = F_S(y|\boldsymbol{X}_S = \boldsymbol{x}_S)$, we have

$$\widehat{F}_S(y|\boldsymbol{X}_S = \boldsymbol{x}_S, \Theta_1, \dots, \Theta_k, \mathcal{D}_n) = \sum_{i=1}^n w_{n,i}(\boldsymbol{x}_S) \mathbb{1}_{\{U_i \leq s_i\}}$$
$$= \sum_{i=1}^n w_{n,i}(\boldsymbol{x}_S) \mathbb{1}_{\{\tilde{U}_i \leq s\}},$$

where $\tilde{U}_i \sim \mathcal{U}(s_i, s_i + 1), i = 1, \dots, n$ are independent uniform variable. Then, A.4 is equivalent to:

$$\forall \boldsymbol{x} \in \mathbb{R}^d, \forall s \in [0, 1], \quad \sum_{i=1}^n w_{n,i}(\boldsymbol{x}_S) \mathbb{1}_{\{\tilde{U}_i \leq s\}} \xrightarrow[n \to +\infty]{a.s} s \quad \text{(A.5)}$$

A.5 states that, $\forall s \in [0, 1], \exists N_s \subset \Omega, \mathbb{P}(N_s) = 0$ such that

$$\forall \omega \in N_s^c, \quad \sum_{i=1}^n w_{n,i}(\boldsymbol{x}_S) \mathbb{1}_{\{\tilde{U}_i(\omega) \leq s\}} \xrightarrow[n \to +\infty]{} s. \quad \text{(A.6)}$$

Thus, we need to find a set $N$ that does not depend on $s$ which satisfies A.6 to get the uniform convergence with Dini's second theorem. To that aim, we will use the density of $\mathbb{Q}$ in $\mathbb{R}$ as in the proof of the Glivenko-Cantelli theorem.

Since the countable union of null set is a null set, $\exists N \subset \Omega, \mathbb{P}(N) = 0$ such that

$$\forall s \in [0,1] \cap \mathbb{Q}, \ \forall \omega \in N^c, \quad \sum_{i=1}^{n} w_{n,i}(\boldsymbol{x}_S) \mathbb{1}_{\{\tilde{U}_i(\omega) \leq s\}} \xrightarrow[n \to +\infty]{} s \tag{A.7}$$

A.7 is also true $\forall s \in [0,1]$. Indeed, let $s \in [0,1], \epsilon > 0, w \in N, \exists p, q \in \mathbb{Q}$ such that $s - \epsilon \leq p \leq s \leq q \leq s + \epsilon$, since $s \to \sum_{i=1}^{n} w_{n,i}(\boldsymbol{x}_S) \mathbb{1}_{\{\tilde{U}_i(w) \leq s\}}$ is increasing, we have:

$$\sum_{i=1}^{n} w_{n,i}(\boldsymbol{x}_S) \mathbb{1}_{\{\tilde{U}_i(\omega) \leq p\}} \leq \sum_{i=1}^{n} w_{n,i}(\boldsymbol{x}_S) \mathbb{1}_{\{\tilde{U}_i(\omega) \leq s\}} \leq \sum_{i=1}^{n} w_{n,i}(\boldsymbol{x}_S) \mathbb{1}_{\{\tilde{U}_i(\omega) \leq q\}} \tag{A.8}$$

Thus,

$$s - \epsilon \leq \liminf \sum_{i=1}^{n} w_{n,i}(\boldsymbol{x}_S) \mathbb{1}_{\{\tilde{U}_i(\omega) \leq s\}} \leq \limsup \sum_{i=1}^{n} w_{n,i}(\boldsymbol{x}_S) \mathbb{1}_{\{\tilde{U}_i(\omega) \leq s\}} \leq s + \epsilon \tag{A.9}$$

So we have shown that $\exists N \subset \Omega, \mathbb{P}(N) = 0, \forall \omega \in N^c$

- $s \to \sum_{i=1}^{n} w_{n,i}(\boldsymbol{x}_S) \mathbb{1}_{\{\tilde{U}_i(w) \leq s\}}$ is increasing for all $n \in \mathbb{N}^\star$
- $\forall s \in [0,1], \quad \sum_{i=1}^{n} w_{n,i}(\boldsymbol{x}_S) \mathbb{1}_{\{\tilde{U}_i(w) \leq s\}} \xrightarrow[n \to +\infty]{} s$ and $s \to s$ is continuous

Then the Dini's second theorem states that we have the almost sure uniform convergence proving Theorem 4.4. Now, we turn to the proof of Proposition A.5 and Lemma A.1. To that aim, we need the following lemma based on Vapnik-Chervonenkis classes.

**Lemma A.2.** *Consider $\mathcal{D}_n, \mathcal{D}_n^\diamond$, two independent datasets of independent n samples of $(\boldsymbol{X}, Y)$. Build a tree using $\mathcal{D}_n$ with bootstrap and bagging procedure driven by $\Theta$. As before, $N(A_n^{(S)}(\boldsymbol{x}_S; \Theta_l))$ is the number of bootstrap observations of $\mathcal{D}_n$ that fall into $A_n^{(S)}(\boldsymbol{x}_S; \Theta_l, \mathcal{D}_n)$ and $N^\diamond(A_n^{(S)}(\boldsymbol{x}_S; \Theta_l))$ is the number of observations of $\mathcal{D}_n^\diamond$ that fall into $A_n^{(S)}(\boldsymbol{x}_S; \Theta_l, \mathcal{D}_n)$. Then:*

$$\forall \epsilon > 0, \quad \mathbb{P}\left(\left|N(A_n^{(S)}(\boldsymbol{x}_S; \Theta_l)) - N^\diamond(A_n^{(S)}(\boldsymbol{x}_S; \Theta_l))\right| > \epsilon\right) \leq 24(n+1)^{2|S|} e^{-\epsilon^2/288n} \tag{A.10}$$

See the proof in [Elie-Dit-Cosaque and Maume-Deschamps, 2020], Lemma 5.3.

*Proof of proposition A.5.*

We want to show that:

$$\forall \boldsymbol{x} \in \mathbb{R}^d, \forall y \in \mathbb{R}, \quad F_S^\diamond(y|\boldsymbol{X}_S = \boldsymbol{x}_S, \Theta_1, \ldots, \Theta_k, \mathcal{D}_n, \mathcal{D}_n^\diamond) \xrightarrow[n \to +\infty]{a.s} F_S(y|\boldsymbol{X}_S = \boldsymbol{x}_S)|$$

Let $x \in \mathbb{R}^d, y \in \mathbb{R}$, we have:

$$|F_S^\diamond(y|\boldsymbol{x}_S) - F_S(y|\boldsymbol{X}_S = \boldsymbol{x}_S)| \leq \left|\sum_{i=1}^{n} w_{n,i}^\diamond(\boldsymbol{x}_S)\left(\mathbb{1}_{\{Y^{\diamond i} \leq y\}} - F_S(y|\boldsymbol{X}_S^{\diamond i})\right)\right|$$

$$+ \left|\sum_{i=1}^{n} w_{n,i}^\diamond(\boldsymbol{x}_S)\left(F_S(y|\boldsymbol{X}_S^{\diamond i}) - F_S(y|\boldsymbol{X}_S = \boldsymbol{x}_S)\right)\right|$$

We define $W_n = \sum_{i=1}^{n} w_{n,i}^\diamond(\boldsymbol{x}_S)\left(\mathbb{1}_{\{Y^{\diamond i} \leq y\}} - F_S(y|\boldsymbol{X}_S^{\diamond i})\right) =$ and $V_n = \sum_{i=1}^{n} w_{n,i}^\diamond(\boldsymbol{x}_S)\left(F_S(y|\boldsymbol{X}_S^{\diamond i}) - F_S(y|\boldsymbol{X}_S = \boldsymbol{x}_S)\right)$ and treat each term separately.

Let prove that $|W_n| \xrightarrow[n \to +\infty]{a.s.} 0$. We can rewrite $W_n$ as $W_n = \sum_{i=1}^{n} w_{n,i}^{\diamond}(\boldsymbol{x}_S) H_i^{\diamond}$ where $H_i^{\diamond}$ is bounded by 1 and $E[H_i^{\diamond}|X_S^{\diamond i}] = 0$. Then,

$$
\begin{aligned}
P(W_n > \epsilon) &\leq e^{-t\epsilon} \, \mathbb{E}[e^{tW_n}] \\
&\leq e^{-t\epsilon} \, \mathbb{E}\left[\prod_{i=1}^{n} \mathbb{E}\left[e^{tw_{n,i}^{\diamond}(\boldsymbol{x}_S)H^{\diamond i}}|\Theta_1, \ldots, \Theta_k, \mathcal{D}_n, X_S^{\diamond,i}, \ldots, X_S^{\diamond,n}\right]\right] \\
&\leq e^{-t\epsilon} \, \mathbb{E}\left[\prod_{i=1}^{n} e^{\frac{t^2}{2} w_{n,i}^{\diamond}(\boldsymbol{x}_S)^2}\right]
\end{aligned}
$$

The last inequality comes from the fact that $w_{n,i}^{\diamond}(\boldsymbol{x}_S)$ is a constant given $\Theta_1, \ldots, \Theta_k, \mathcal{D}_n, X_S^{\diamond,i}, \ldots, X_S^{\diamond,n}$, and as $H^{\diamond i}$ is bounded by 1 with $E[H_i^{\diamond}|X_S^{\diamond i}] = 0$, we used the following inequality: If $|X| \leq 1$ a.s and $\mathbb{E}[X] = 0$, then $\mathbb{E}[e^{tX}] \leq \mathbb{E}[e^{\frac{t^2}{2}}]$.

By using Assumption A.2, item 2., let $K > 0$ be such that $\forall l \in [k]$, $N(A_n^{(S)}(\boldsymbol{x}_S; \Theta_l)) \geq \frac{K\sqrt{n}\ln(n)^{\beta}}{2}$ a.s., then we have $\Gamma(l) = \{N^{\diamond}(A_n^{(S)}(\boldsymbol{x}_S; \Theta_l)) \leq \frac{K\sqrt{n}\ln(n)^{\beta}}{2}\} \subset \{|N(A_n^{(S)}(\boldsymbol{x}_S; \Theta_l)) - N^{\diamond}(A_n^{(S)}(\boldsymbol{x}_S; \Theta_l))| \geq \frac{K\sqrt{n}\ln(n)^{\beta}}{2}\}$. Thus, using Lemma A.2, we have that $\mathbb{P}(\Gamma(l)) \leq 24(n + 1)^{2|S|} \exp(-\frac{-K^2(\ln(n)^{2\beta})}{1152})$.

We have

$$
\begin{aligned}
\sum_{i=1}^{n} w_{n,i}^{\diamond}(\boldsymbol{x}_S)^2 &= \sum_{i=1}^{n} \frac{w_{n,i}^{\diamond}(\boldsymbol{x}_S)}{k}\left(\sum_{l=1}^{k} \frac{\mathbb{1}_{X^{\diamond i} \in A_n^{(S)}(\boldsymbol{x}_S; \Theta_l, \mathcal{D}_n)}}{N^{\diamond}(A_n^{(S)}(\boldsymbol{x}_S; \Theta_l))}(\mathbb{1}_{\{\Gamma(l)\}} + \mathbb{1}_{\{\Gamma(l)^c\}})\right) \\
&\leq \sum_{i=1}^{n} w_{n,i}^{\diamond}(\boldsymbol{x}_S)\left(\frac{2}{K\sqrt{n}\ln(n)^{\beta}} + \frac{1}{k}\sum_{l=1}^{k} \mathbb{1}_{X^{\diamond i} \in A_n^{(S)}(\boldsymbol{x}_S; \Theta_l, \mathcal{D}_n)}\mathbb{1}_{\{\Gamma(l)\}}\right)
\end{aligned}
$$

So that,

$$
\begin{aligned}
P(W_n > \epsilon) &\leq \exp(-t\epsilon + \frac{t^2}{K\sqrt{n}\ln(n)^{\beta}})\mathbb{E}\left[\exp\left(\frac{t^2}{2}\mathbb{1}_{\cup_{l=1}^{k}\Gamma(l)}\right)\right] \\
&\leq \exp(-t\epsilon + \frac{t^2}{K\sqrt{n}\ln(n)^{\beta}}) \times \left(1 + e^{\frac{t^2}{2}}\sum_{l=1}^{k}\mathbb{P}(\Gamma(l))\right) \\
&\leq \exp(-t\epsilon + \frac{t^2}{K\sqrt{n}\ln(n)^{\beta}}) \times \left(1 + 24k(n+1)^{2|S|}\exp\left(\frac{t^2}{2} - \frac{K^2\ln(n)^{2\beta}}{1152}\right)\right)
\end{aligned}
$$

Taking $t^2 = \frac{K^2\ln(n)^{2\beta}}{576}$ leads to

$$
P(W_n > \epsilon) \leq (1 + 24k(n+1)^{2|S|})\exp\left(\frac{K\ln(n)^{\beta}}{576\sqrt{n}} - \frac{\epsilon K\ln(n)^{\beta}}{24}\right)
$$

We obtain the same bound for $\mathbb{P}(W_n \leq -\epsilon) = \mathbb{P}(-W_n > \epsilon)$, then by using Assumption A.2, item 1., $k = \mathcal{O}(n^{\alpha})$ so that the right term is finite, we conclude by Borel cantelli that $|W_n|$ goes to 0 a.s.

Finally, we show that $V_n \xrightarrow[n \to +\infty]{a.s.} 0$.

$$|V_n| = \left| \sum_{i=1}^{n} w_{n,i}^{\diamond}(\boldsymbol{x}_S)\Big(F_S(y|\boldsymbol{X}_S^{\diamond i}) - F_S(y|\boldsymbol{X}_S = \boldsymbol{x}_S)\Big) \right|$$

$$\leq \sum_{i=1}^{n} \sum_{l=1}^{k} \frac{\mathbb{1}_{X^{\diamond i} \in A_n^{(S)}(\boldsymbol{x}_S, \Theta_l, \mathcal{D}_n)}}{N^{\diamond}(A_n^{(S)}(\boldsymbol{x}_S, \Theta_l))} \left| \Big(F_S(y|\boldsymbol{X}_S^{\diamond i}) - F_S(y|\boldsymbol{X}_S = \boldsymbol{x}_S)\Big) \right|$$

$$\leq \sum_{l=1}^{k} \left( \sum_{i=1}^{n} \frac{\mathbb{1}_{X^{\diamond i} \in A_n^{(S)}(\boldsymbol{x}_S, \Theta_l, \mathcal{D}_n)}}{N^{\diamond}(A_n^{(S)}(\boldsymbol{x}_S, \Theta_l))} \left| F_S(y|\boldsymbol{X}_S^{\diamond i}) - F_S(y|\boldsymbol{X}_S = \boldsymbol{x}_S) \right| \right)$$

$$\leq \sum_{l=1}^{k} \left( \sum_{i=1}^{n} \frac{\mathbb{1}_{X^{\diamond i} \in A_n^{(S)}(\boldsymbol{x}_S, \Theta_l, \mathcal{D}_n)}}{N^{\diamond}(A_n^{(S)}(\boldsymbol{x}_S, \Theta_l))} \sup_{\boldsymbol{z} \in A_n^{(S)}(\boldsymbol{x}; \Theta_l, \mathcal{D}_n)} |F_S(y|\boldsymbol{X}_S = z_S) - F_S(y|\boldsymbol{X}_S = \boldsymbol{x}_S)| \right)$$

$$\leq \sum_{l=1}^{k} \sup_{\boldsymbol{z} \in A_n^{(S)}(\boldsymbol{x}; \Theta_l, \mathcal{D}_n)} |F_S(y|\boldsymbol{X}_S = z_S) - F_S(y|\boldsymbol{X}_S = \boldsymbol{x}_S)|$$

By assumption A.1 i.e that variation of the Projected CDF within the cell of the Projected Tree vanishes, we conclude that $|V_n| \xrightarrow[n \to +\infty]{a.s} 0$ ending the proof of Proposition A.5.

*Proof of Lemma A.1.*

We want to show that:

$$\forall \boldsymbol{x} \in \mathbb{R}^d, \forall y \in \mathbb{R} \quad |F_S^{\diamond}(y|\boldsymbol{X}_S = \boldsymbol{x}_S, \Theta_1, \dots, \Theta_k, \mathcal{D}_n) - \widehat{F}_S(y|\boldsymbol{X}_S = \boldsymbol{x}_S, \Theta_1, \dots, \Theta_k, \mathcal{D}_n| \xrightarrow{a.s} 0.$$

We have

$$|F_S^{\diamond}(y|\boldsymbol{X}_S = \boldsymbol{x}_S) - \widehat{F}_S(y|\boldsymbol{X}_S = \boldsymbol{x}_S)| = |\sum_{i=1}^{n} w_{n,i}^{\diamond}(\boldsymbol{x}_S)\mathbb{1}_{\{Y^{\diamond i} \leq y\}} - w_{n,i}(\boldsymbol{x}_S)\mathbb{1}_{\{Y^i \leq y\}}|$$

$$= \left| \sum_{i=1}^{n} \frac{1}{k} \sum_{l=1}^{k} \frac{\mathbb{1}_{X^{\diamond i} \in A_n^{(S)}(\boldsymbol{x}_S, \Theta_l, \mathcal{D}_n)}}{N^{\diamond}(A_n^{(S)}(\boldsymbol{x}_S, \Theta_l))}\mathbb{1}_{\{Y^{\diamond i} \leq y\}} - \frac{1}{k} \sum_{l=1}^{k} \frac{B_n(X^i; \Theta_l)\mathbb{1}_{X^i \in A_n^{(S)}(\boldsymbol{x}_S, \Theta_l, \mathcal{D}_n)}}{N(A_n^{(S)}(\boldsymbol{x}_S, \Theta_l))}\mathbb{1}_{\{Y^i \leq y\}} \right|$$

$$= \left| \frac{1}{k} \sum_{l=1}^{k} \left( \frac{\sum_{i=1}^{n} \mathbb{1}_{X^{\diamond i} \in A_n^{(S)}(\boldsymbol{x}_S, \Theta_l, \mathcal{D}_n)}\mathbb{1}_{\{Y^{\diamond i} \leq y\}}}{N^{\diamond}(A_n^{(S)}(\boldsymbol{x}_S, \Theta_l))} - \frac{\sum_{i=1}^{n} B_n(X^i; \Theta_l)\mathbb{1}_{X^i \in A_n^{(S)}(\boldsymbol{x}_S, \Theta_l, \mathcal{D}_n)}\mathbb{1}_{\{Y^i \leq y\}}}{N(A_n^{(S)}(\boldsymbol{x}_S, \Theta_l))} \right) \right|$$

As in [Arenal-Gutiérrez et al., 1996], we replace the boostrap component with $Z^1, \dots, Z^n$ which are distributed as $Z = (Z_1, Z_2)$ that has uniform distribution over $\mathcal{D}_n = \{(\boldsymbol{X}^1, Y^1), \dots, (\boldsymbol{X}^n, Y^n)\}$ conditionally to $\mathcal{D}_n$.

$$|F_S^{\diamond}(y|\boldsymbol{X}_S = \boldsymbol{x}_S) - \widehat{F}_S(y|\boldsymbol{X}_S = \boldsymbol{x}_S)|$$

$$= \left| \frac{1}{k} \sum_{l=1}^{k} \left( \frac{\sum_{i=1}^{n} \mathbb{1}_{\{X^{\diamond i} \in A_n^{(S)}(\boldsymbol{x}_S, \Theta_l, \mathcal{D}_n), Y^{\diamond i} \leq y\}}}{N^{\diamond}(A_n^{(S)}(\boldsymbol{x}_S, \Theta_l))} - \frac{\sum_{i=1}^{n} \mathbb{1}_{\{Z_1^i \in A_n^{(S)}(\boldsymbol{x}_S, \Theta_l, \mathcal{D}_n), Z_2^i \leq y\}}}{N(A_n^{(S)}(\boldsymbol{x}_S, \Theta_l))} \right) \right|$$

$$\leq \frac{1}{k} \sum_{l=1}^{k} \left| \frac{\sum_{i=1}^{n} \mathbb{1}_{\{X^{\diamond i} \in A_n^{(S)}(\boldsymbol{x}_S, \Theta_l, \mathcal{D}_n), Y^{\diamond i} \leq y\}}}{N^{\diamond}(A_n^{(S)}(\boldsymbol{x}_S, \Theta_l))} - \frac{\sum_{i=1}^{n} \mathbb{1}_{\{Z_1^i \in A_n^{(S)}(\boldsymbol{x}_S, \Theta_l, \mathcal{D}_n), Z_2^i \leq y\}}}{N(A_n^{(S)}(\boldsymbol{x}_S, \Theta_l))} \right|$$

$$\stackrel{\text{def}}{=} \frac{1}{k} \sum_{l=1}^{k} |G_l|$$

We have,

$$|G_l| \leq |G_l^1| + |G_l^2|$$

with

$$G_l^1 = \frac{\left| \sum_{i=1}^n \mathbb{1}_{\{X^{\diamond i} \in A_n^{(S)}(\boldsymbol{x}_S, \Theta_l, \mathcal{D}_n),\ Y^{\diamond i} \leq y\}} - \sum_{i=1}^n \mathbb{1}_{\{Z_1^i \in A_n^{(S)}(\boldsymbol{x}_S, \Theta_l, \mathcal{D}_n),\ Z_2^i \leq y\}} \right|}{N(A_n^{(S)}(\boldsymbol{x}_S, \Theta_l))} \text{ and}$$

$$G_l^2 = \frac{\left| \sum_{i=1}^n \mathbb{1}_{\{X^{\diamond i} \in A_n^{(S)}(\boldsymbol{x}_S, \Theta_l, \mathcal{D}_n),\ Y^{\diamond i} \leq y\}} - \sum_{i=1}^n \mathbb{1}_{\{Z_1^i \in A_n^{(S)}(\boldsymbol{x}_S, \Theta_l, \mathcal{D}_n),\ Z_2^i \leq y\}} \right|}{N^{\diamond}(A_n^{(S)}(\boldsymbol{x}_S, \Theta_l))}.$$

Therefore, A.1 is equivalent to show that $\forall l \in [k]$, $|G_l^1|, |G_l^2| \xrightarrow[n \to +\infty]{a.s} 0$.

Let $\epsilon > 0$, by using Assumption A.2, item 2. i.e $\exists K > 0, N(A_n^{(S)}(\boldsymbol{x}_S, \Theta_l)) \geq K\sqrt{n}\ln(n)^\beta$, we have,

$$\mathbb{P}(|G_l^1| > \epsilon) = \mathbb{P}\left[ \left| \frac{1}{n} \sum_{i=1}^n \mathbb{1}_{\{X^{\diamond i} \in A_n^{(S)}(\boldsymbol{x}_S, \Theta_l, \mathcal{D}_n),\ Y^{\diamond i} \leq y\}} - \frac{1}{n} \sum_{i=1}^n \mathbb{1}_{\{Z_1^i \in A_n^{(S)}(\boldsymbol{x}_S, \Theta_l, \mathcal{D}_n),\ Z_2^i \leq y\}} \right| > \frac{\epsilon N(A_n^{(S)}(\boldsymbol{x}_S, \Theta_l))}{n} \right]$$

$$\leq \mathbb{P}\left[ \sup_{A \in \mathcal{B}} \left| \frac{1}{n} \sum_{i=1}^n \mathbb{1}_{\{(\boldsymbol{X}^{\diamond i}, Y^{\diamond i}) \in A\}} - \mathbb{P}((\boldsymbol{X}, Y) \in A) \right| > \frac{\epsilon\, K\sqrt{n}\ln(n)^\beta}{3n} \right]$$

$$+ \mathbb{P}\left[ \sup_{A \in \mathcal{B}} \left| \frac{1}{n} \sum_{i=1}^n \mathbb{1}_{\{(\boldsymbol{X}^i, Y^i) \in A\}} - \mathbb{P}((\boldsymbol{X}, Y) \in A) \right| > \frac{\epsilon\, K\sqrt{n}\ln(n)^\beta}{3n} \right]$$

$$+ \mathbb{P}\left[ \sup_{A \in \mathcal{B}} \left| \frac{1}{n} \sum_{i=1}^n \mathbb{1}_{\{(\boldsymbol{Z}_1^i, \boldsymbol{Z}_2^i) \in A\}} - \sum_{i=1}^n \mathbb{1}_{\{(\boldsymbol{X}^i, Y^i) \in A\}} \right| > \frac{\epsilon\, K\sqrt{n}\ln(n)^\beta}{3n} \right]$$

$$\leq 2\mathbb{P}\left[ \sup_{A \in \mathcal{B}} \left| \frac{1}{n} \sum_{i=1}^n \mathbb{1}_{\{(\boldsymbol{X}^i, Y^i) \in A\}} - \mathbb{P}((\boldsymbol{X}, Y) \in A) \right| > \frac{\epsilon\, K\sqrt{n}\ln(n)^\beta}{3n} \right]$$

$$+ \mathbb{P}\left[ \sup_{A \in \mathcal{B}} \left| \frac{1}{n} \sum_{i=1}^n \mathbb{1}_{\{(\boldsymbol{Z}_1^i, \boldsymbol{Z}_2^i) \in A\}} - \sum_{i=1}^n \mathbb{1}_{\{(\boldsymbol{X}^i, Y^i) \in A\}} \right| > \frac{\epsilon\, K\sqrt{n}\ln(n)^\beta}{3n} \right]$$

where $\mathcal{B} = \left\{ \prod_{i=1}^{|S|}[a_i, b_i] \times [-\infty, y] : a_i, b_i \in \bar{\mathbb{R}} \right\}$. The first term are handled thanks to a direct application of the Theorem 2 in [Vapnik, 1971] that bounds the difference between the frequencies of events to their probabilities over an entire class $\mathcal{B}$. Therefore,

$$\mathbb{P}\left[ \sup_{A \in \mathcal{B}} \left| \frac{1}{n} \sum_{i=1}^n \mathbb{1}_{\{(\boldsymbol{X}^i, Y^i) \in A\}} - \mathbb{P}((\boldsymbol{X}, Y) \in A) \right| > \frac{\epsilon\, K\sqrt{n}\ln(n)^\beta}{3n} \right] \leq 8(n+1)^{2|S|+1} \exp\left(-\frac{K^2\epsilon^2 \ln(n)^{2\beta}}{288}\right)$$

To handle the last term, we apply the Theorem 2 in [Vapnik, 1971] under the conditional distribution given $\mathcal{D}_n$,

$$\mathbb{P}\left[ \sup_{A \in \mathcal{B}} \left| \frac{1}{n} \sum_{i=1}^n \mathbb{1}_{\{(\boldsymbol{Z}_1^i, \boldsymbol{Z}_2^i) \in A\}} - \sum_{i=1}^n \mathbb{1}_{\{(\boldsymbol{X}^i, Y^i) \in A\}} \right| > \frac{\epsilon\, K\sqrt{n}\ln(n)^\beta}{3n} \right]$$

$$= \mathbb{E}\left[ \mathbb{P}\left[ \sup_{A \in \mathcal{B}} \left| \frac{1}{n} \sum_{i=1}^n \mathbb{1}_{\{(\boldsymbol{Z}_1^i, \boldsymbol{Z}_2^i) \in A\}} - \sum_{i=1}^n \mathbb{1}_{\{(\boldsymbol{X}^i, Y^i) \in A\}} \right| > \frac{\epsilon\, K\sqrt{n}\ln(n)^\beta}{3n} \,\Big|\, \mathcal{D}_n \right] \right]$$

$$= \mathbb{E}\left[ \mathbb{P}\left[ \sup_{A \in \mathcal{B}} \left| \frac{1}{n} \sum_{i=1}^n \mathbb{1}_{\{(\boldsymbol{Z}_1^i, \boldsymbol{Z}_2^i) \in A\}} - \mathbb{P}((\boldsymbol{Z}_1, Z_2) \in A \,|\, D_n) \right| > \frac{\epsilon\, K\sqrt{n}\ln(n)^\beta}{3n} \,\Big|\, \mathcal{D}_n \right] \right]$$

$$\leq 8(n+1)^{2|S|+1} \exp\left(-\frac{K^2\epsilon^2 \ln(n)^{2\beta}}{288}\right)$$

Finally, we get the overall upper bound,

$$\mathbb{P}(|G_l^1| > \epsilon) \leq 24(n+1)^{2|S|+1} \exp\left(-\frac{\epsilon^2 \ln(n)^2 \beta}{288}\right)$$

By Borel-Cantelli, we conclude that $|G_l^1| \xrightarrow[n \to +\infty]{a.s} 0$.

Now, we treat the term $|G_l^2|$. The main difference with $|G_l^1|$ is that $N(A_n^{(S)}(\boldsymbol{x}_S, \Theta_l))$ was replaced by $N^\diamond(A_n^{(S)}(\boldsymbol{x}_S, \Theta_l))$. The proof remains the same as above, we just have to show how to deal with $N^\diamond(A_n^{(S)}(\boldsymbol{x}_S, \Theta_l))$ for each term. For example,

$$\mathbb{P}\left[\sup_{A \in \mathcal{B}} \left|\frac{1}{n} \sum_{i=1}^n \mathbb{1}_{\{(\boldsymbol{Z}_1^i, \boldsymbol{Z}_2^i) \in A\}} - \sum_{i=1}^n \mathbb{1}_{\{(\boldsymbol{X}^i, Y^i) \in A\}}\right| > \frac{\epsilon\, N^\diamond(A_n^{(S)}(\boldsymbol{x}_S, \Theta_l))}{3n}\right]$$

$$= \mathbb{P}\left[\sup_{A \in \mathcal{B}} \left|\frac{1}{n} \sum_{i=1}^n \mathbb{1}_{\{(\boldsymbol{Z}_1^i, \boldsymbol{Z}_2^i) \in A\}} - \sum_{i=1}^n \mathbb{1}_{\{(\boldsymbol{X}^i, Y^i) \in A\}}\right| > \frac{\epsilon\, N^\diamond(A_n^{(S)}(\boldsymbol{x}_S, \Theta_l))}{3n}, \; \exists l \in [k], |N^\diamond(A_n^{(S)}(\boldsymbol{x}_S, \Theta_l)) - N(A_n^{(S)}(\boldsymbol{x}_S, \Theta_l))| > \lambda\right]$$

$$+ \mathbb{P}\left[\sup_{A \in \mathcal{B}} \left|\frac{1}{n} \sum_{i=1}^n \mathbb{1}_{\{(\boldsymbol{Z}_1^i, \boldsymbol{Z}_2^i) \in A\}} - \sum_{i=1}^n \mathbb{1}_{\{(\boldsymbol{X}^i, Y^i) \in A\}}\right| > \frac{\epsilon\, N^\diamond(A_n^{(S)}(\boldsymbol{x}_S, \Theta_l))}{3n}, \; \forall l \in [k], |N^\diamond(A_n^{(S)}(\boldsymbol{x}_S, \Theta_l)) - N(A_n^{(S)}(\boldsymbol{x}_S, \Theta_l))| \le \lambda\right]$$

$$\le k\, \mathbb{P}\left(|N^\diamond(A_n^{(S)}(\boldsymbol{x}_S, \Theta_l)) - N(A_n^{(S)}(\boldsymbol{x}_S, \Theta_l))| > \lambda\right)$$

$$+ \mathbb{P}\left[\sup_{A \in \mathcal{B}} \left|\frac{1}{n} \sum_{i=1}^n \mathbb{1}_{\{(\boldsymbol{Z}_1^i, \boldsymbol{Z}_2^i) \in A\}} - \sum_{i=1}^n \mathbb{1}_{\{(\boldsymbol{X}^i, Y^i) \in A\}}\right| > \frac{\epsilon\, (N(A_n^{(S)}(\boldsymbol{x}_S, \Theta_l)) - \lambda)}{3n}\right]$$

By using Lemma A.2 for the first term, and Assumption A.2, item 2. with $\lambda = 2K\sqrt{n}\ln(n)^\beta$, we have

$$\mathbb{P}\left[\sup_{A \in \mathcal{B}} \left|\frac{1}{n} \sum_{i=1}^n \mathbb{1}_{\{(\boldsymbol{Z}_1^i, \boldsymbol{Z}_2^i) \in A\}} - \sum_{i=1}^n \mathbb{1}_{\{(\boldsymbol{X}^i, Y^i) \in A\}}\right| > \frac{\epsilon\, N^\diamond(A_n^{(S)}(\boldsymbol{x}_S, \Theta_l))}{3n}\right]$$

$$\le k\, 24(n+1)^{2|S|} e^{-\epsilon^2/288n} + 24(n+1)^{2|S|+1} \exp\left(-\frac{K^2 \epsilon^2 \ln(n)^{2\beta}}{288}\right)$$

Since $k = \mathcal{O}(n^\alpha)$ by Assumption A.2, item 1., the right hand is summable, then we conclude that $|G_l^2| \xrightarrow[n \to +\infty]{a.s} 0$.

This conclude the proof of Lemma A.1, thus the proof of Theorem 4.4.

# B  Empirical evaluations of the estimator $\widehat{F}_S$

In order to compare the PRF CDF $\widehat{F}_S(y|\boldsymbol{X}_S = \boldsymbol{x}_S)$ and $F_S(y|\boldsymbol{X}_S = \boldsymbol{x}_S)$, we use a Monte Carlo estimator to effectively compute $F_S(y|\boldsymbol{X}_S = \boldsymbol{x}_S)$. We use the synthetic dataset of Section 5: $\boldsymbol{X} \in \mathbb{R}^p$, $\boldsymbol{X} \in \mathcal{N}(0, \Sigma)$, $\Sigma = 0.8 J_p + 5 I_p$ with $p = 100$, $I_p$ is the identity matrix, $J_p$ is all-ones matrix and a linear predictor with switch defined as:

$$Y = (X_1 + X_2)\mathbb{1}_{X_5 \le 0} + (X_3 + X_4)\mathbb{1}_{X_5 > 0}. \tag{B.1}$$

The variables $X_i$ for $i = 6 \ldots 100$ are noise variables. We fit a RF with a sample size $n = 10^4$, $k = 20$ trees and the minimal number of samples by leaf node is set to $t_n = \lfloor \sqrt{n} \times \ln(n)^{1.5}/250 \rfloor$ for the original and the Projected Forest.

We chose a randomly chosen point $\boldsymbol{x}_S = [-0.13, 1.29, -1.31]$ with $S = [1, 2, 5]$ from the test set. The experiment is replicated 100 times. Figure 1 shows that the estimator works well for almost all points $y \in \mathbb{R}$.

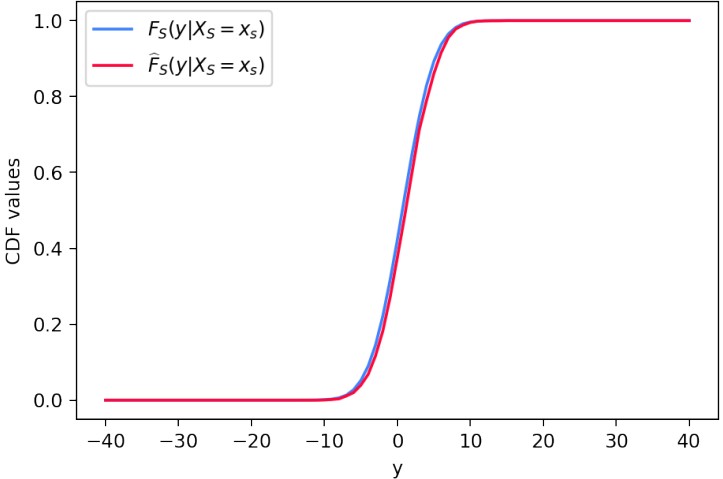

Figure 1: Comparison of $\widehat{F}_S(y|\boldsymbol{X}_S = \boldsymbol{x}_S)$ and $F_S(y|\boldsymbol{X}_S = \boldsymbol{x}_S)$ with $S = [1, 2, 5]$ and $\boldsymbol{x}_S = [-0.13, 1.29, -1.31]$

We also compute two global metrics. For a given $S$, we compute the average Kolmogorov-Smirnov $MKS = \frac{1}{n}\sum_{i=1}^{n} \sup_{y \in \mathbb{R}} |\widehat{F}_S(y|\boldsymbol{X}_S = \boldsymbol{x}_{S,i}) - F_S(y|\boldsymbol{X}_S = \boldsymbol{x}_{S,i})|$ and the average mean absolute deviation $MAD = \frac{1}{n}\sum_{i=1}^{n} \int_{\mathbb{R}} |\widehat{F}_S(y|\boldsymbol{X}_S = \boldsymbol{x}_{S,i}) - F_S(y|\boldsymbol{X}_S = \boldsymbol{x}_{S,i})| dy$.

We have MAD = 0.008 and the MKS=0.26 on all the observations with $S = [1, 2, 3, 5]$ showing the estimator's efficiency. We also compute them with small $S = [0, 4]$, it works even better with MAD=0.068, MKS=0.0098.

## C  Additional experiments

### C.1  Local rules of Anchors and Sufficient Rules with ground truth explanations

In this section, we compare Anchors and Sufficient Rules in a synthetic dataset where there are strong dependencies between the important features. In this case, we can evaluate their capacity of providing the ground truth minimal rules since we know the distribution of the data. We use the moon dataset $(X_1, X_2, Y) \in \mathbb{R}^2 \times \{0, 1\}$, see figure 2, and we add gaussian features $\boldsymbol{Z} \in \mathbb{R}^{100}$ with the $\mu, \Sigma$ of the previous section such that the final data is $(X_1, X_2, \boldsymbol{Z}, Y)$. In addition, if $Z_1 > 0$, we flip the label $Y$ of the observations.

We train a RF with the parameters of the previous section. It has AUC=99% on the test set ($10^4$ observations). We use Anchors with threshold $\tau = 0.95$, tolerance $\delta = 0.05$, and the Minimal Sufficient Rules with $\pi = 0.95$ to explain 1000 observations of the test set. We observe that, on

average Anchors tend to give much longer rules. The mean size for Sufficient Rules is 2, and for Anchors it is 10. In addition, the Minimal Sufficient Explanations detect local relevant variables more accurately. It has FDR=3%, TPR=100% and Anchors has FDR=48%, TPR=80%. Finally, we observe qualitatively the rules on a given example $x$ (black star in figure 2). We also test the stability of the explanations by comparing the rules of $x$ and $\tilde{x}$ a nearby observation such that $\max_{i \in \{1,2\}} |x_i - \tilde{x}_i| \leq 0.05$ (yellow star in figure 2). The rules given by Anchors for $x, \tilde{x}$ are $L_{\text{Anchors}}(x) = \{X_1 > -0.03 \text{ AND } Z_1 > 0.01 \text{ AND } Z_9 > -1.66 \text{ AND } Z_{44} > 1.66 \text{ AND } Z_{32} \leq -1.57\}$ and $L_{\text{Anchors}}(\tilde{x}) = \{X_1 > 1.04 \text{ AND } X_2 \leq -0.20 \text{ AND } Z_1 > 0.01 \text{ AND } Z_{28} > 0.01 \text{ AND } Z_{45} \leq -1.57\}$. We find that the rules are very different, showing instability. Moreover, we also note that Anchors is very sensitive to random seed. However, the SDP approach gives the same explanations for $x, \tilde{x}$. The observations have two Minimal Sufficient Explanations $S_1^\star = [x_1, z_1], S_2^\star = [x_2, z_1]$. Thus, they have two Sufficient Rules, we can observe them given the axis $X_1, X_2$ in figure 2. We notice that the rules found are very relevant for explaining these instances' predictions. Nevertheless, the vertical rule could be a little more to the left. We think this inaccuracy comes from the estimation of the RF, which is not perfect.

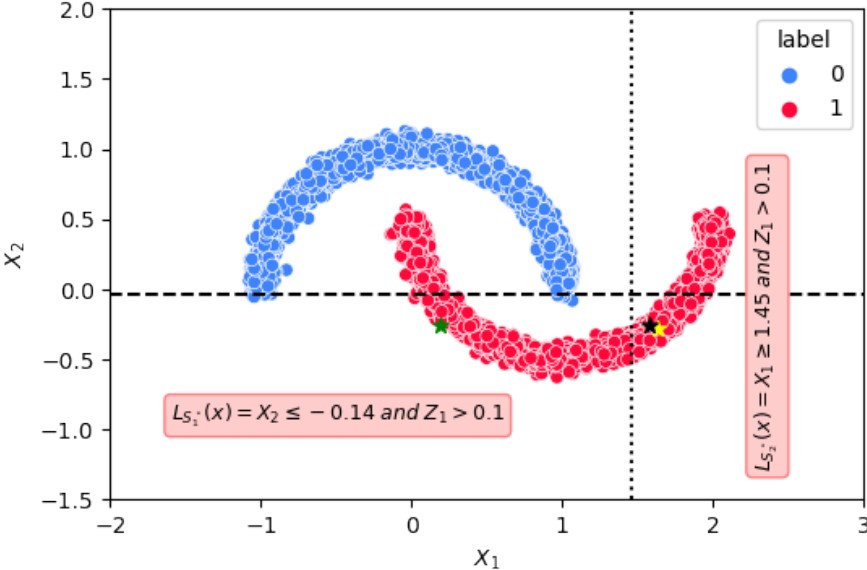

Figure 2: Explanations of $x, \tilde{x}$ by the two Sufficient Rules, the horizontal/vertical rectangle is associate with $S_1^\star = [x_1, z_1], S_2^\star = [x_2, z_1]$ respectively. The background samples are the observations with $z_1 > 0$.

As these observations have multiple explanations, we provide an additional insight about the important variables by computing their Local eXplanatory Importances (LXI). The LXI of $x, \tilde{x}$ are $\begin{bmatrix} x_1 = 0.5, x_2 = 0.5, z_1 = 1, z_2 = 0, \ldots, z_{100} = 0 \end{bmatrix}$. It shows that the variables $\{Z_i\}_{i \in [\![2,100]\!]}$ are irrelevant for these observations. The relevant variables are $X_1, X_2, Z_1$ and especially $Z_1$ is the most important. It is a necessary feature as it appears in every Sufficient Explanations.

**Comparison of SHAP values and LXI on Moon Data:** In figure 3, we compare the SHAP values and LXI of an observation with $Z_1 > 0$ (the green star). We observe that the LXI gives non-null values only on the active variable $(X_2, Z_1)$, while the SV gives non-null values also on noise variables. Moreover, SV gives a non-negligible value to the feature $X_1$ that is not necessary for this prediction. Indeed, by analyzing figure 2, we observe that whatever the value of $X_1$, if we fix $X_2$ and the sign of $Z_1$, the prediction will not change.

We also compute the mean importance score across the population in figure 4. For the SV, we take the mean absolute values as it may have negative contributions. The three important values that come out for both methods are $X_1, X_2, Z_1$. However, as in the local case, SV assign values to the noise variables.

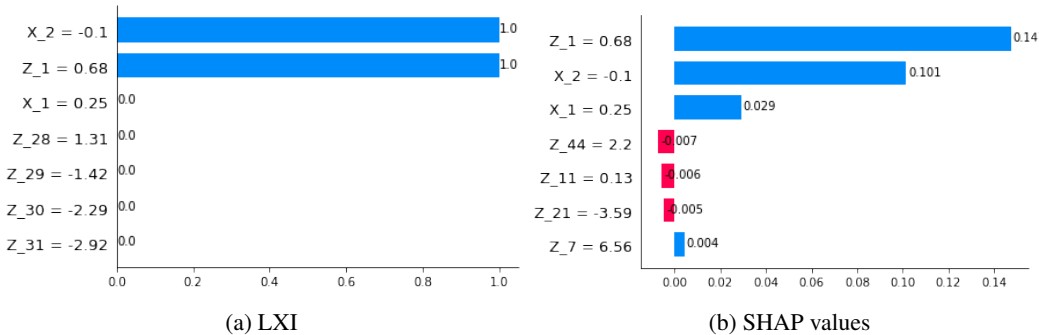

(a) LXI

(b) SHAP values

Figure 3: LXI and SHAP values of the green star

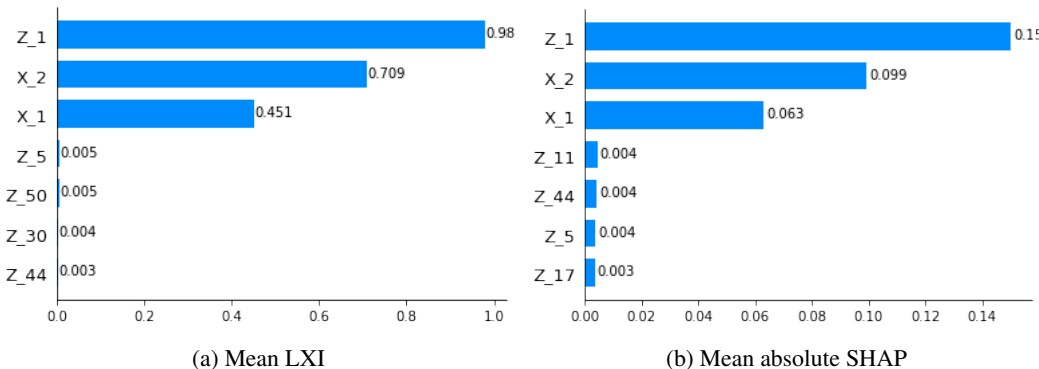

(a) Mean LXI

(b) Mean absolute SHAP

Figure 4: Comparison of Mean LXI and Mean absolute SHAP

## C.2 Shapley values and Local eXplanatory importance (LXI) on LUCAS dataset

In this section, we want to highlight a case where the LXI permit to drastically simplify the different explanations. We use a semi-synthetic dataset LUCAS (LUng CAncer Simple set), a dataset generated by causal Bayesian networks with 12 binary variables. The causal graph is drawn in figure 5 and the probability table in figure 6 .

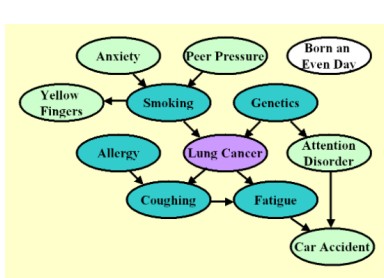

Figure 5: Bayesian network that represents the causal relationships between variables

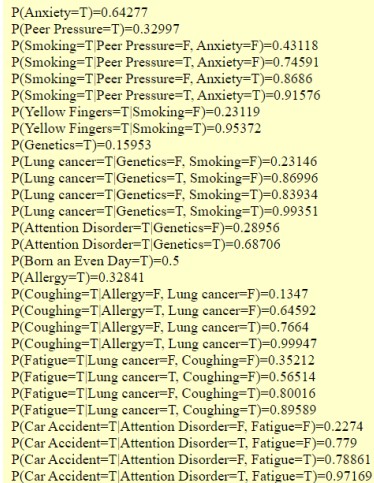

P(Anxiety=T)=0.64277
P(Peer Pressure=T)=0.32997
P(Smoking=T|Peer Pressure=F, Anxiety=F)=0.43118
P(Smoking=T|Peer Pressure=T, Anxiety=F)=0.74591
P(Smoking=T|Peer Pressure=F, Anxiety=T)=0.8686
P(Smoking=T|Peer Pressure=T, Anxiety=T)=0.91576
P(Yellow Fingers=T|Smoking=F)=0.23119
P(Yellow Fingers=T|Smoking=T)=0.95372
P(Genetics=T)=0.15953
P(Lung cancer=T|Genetics=F, Smoking=F)=0.23146
P(Lung cancer=T|Genetics=T, Smoking=F)=0.86996
P(Lung cancer=T|Genetics=F, Smoking=T)=0.83934
P(Lung cancer=T|Genetics=T, Smoking=T)=0.99351
P(Attention Disorder=T|Genetics=F)=0.28956
P(Attention Disorder=T|Genetics=T)=0.68706
P(Born an Even Day=T)=0.5
P(Allergy=T)=0.32841
P(Coughing=T|Allergy=F, Lung cancer=F)=0.1347
P(Coughing=T|Allergy=T, Lung cancer=F)=0.64592
P(Coughing=T|Allergy=F, Lung cancer=T)=0.7664
P(Coughing=T|Allergy=T, Lung cancer=T)=0.99947
P(Fatigue=T|Lung cancer=F, Coughing=F)=0.35212
P(Fatigue=T|Lung cancer=T, Coughing=F)=0.56514
P(Fatigue=T|Lung cancer=F, Coughing=T)=0.80016
P(Fatigue=T|Lung cancer=T, Coughing=T)=0.89589
P(Car Accident=T|Attention Disorder=F, Fatigue=F)=0.2274
P(Car Accident=T|Attention Disorder=T, Fatigue=F)=0.779
P(Car Accident=T|Attention Disorder=F, Fatigue=T)=0.78861
P(Car Accident=T|Attention Disorder=T, Fatigue=T)=0.97169

Figure 6: Probabilities table used to generate Data

In figure 7, we observe the different explanations of an observation chosen randomly, its features values are $\{$**Smoking** $= True$, **Yellow Fingers** $= True$, **Anxiety** $= False$, **Peer Pressure** $= False$, **Genetic** $= False$, **Attention Disorder** $= True$, **Born an Even Day** $= False$, **Car Accident** $= True$, **Fatigue** $= True$, **Allergy** $= False$, **Coughing** $= True\}$ and its label is True. We see in the left of figure 7 that it has many Sufficient Explanations. Therefore, as seen in the right of Figure 7, the LXI permit to synthesize all the different explanations in a single feature contributions that exhibits the local importance of the variables. Each value corresponds to the frequency of apparition of the corresponding feature in the set of all the sufficient explanations.

## All sufficient explanations

| | Sufficient explanations | SDP |
|---|---|---|
| 0 | Smoking - Coughing | 0.9263 |
| 1 | Smoking - Fatigue | 0.9049 |
| 2 | Coughing - Allergy | 0.9245 |
| 3 | Coughing - Fatigue - Yellow_Fingers | 0.9006 |
| 4 | Coughing - Yellow_Fingers - Attention_Disorder | 0.9079 |

## Local Explanatory Importance

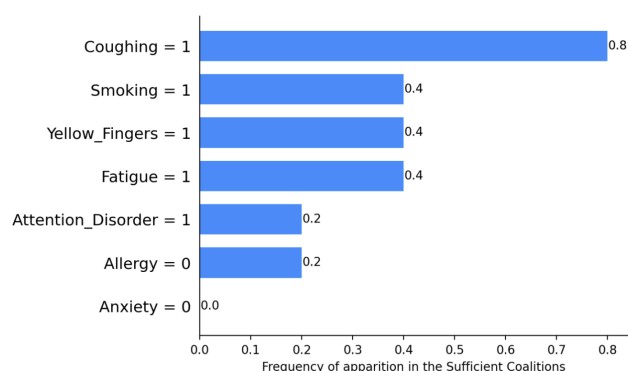

## Feature values highlight by SDP

This observation has 5 different explanations, below to observe their values

Change the explanations

0                                                                                                     ▾

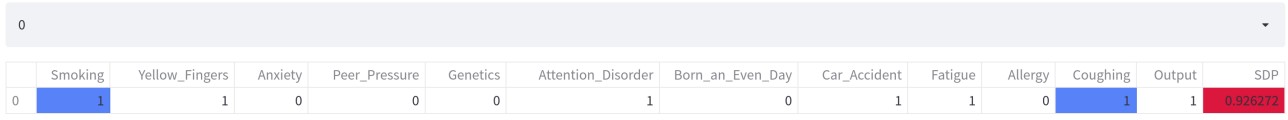

| | Smoking | Yellow_Fingers | Anxiety | Peer_Pressure | Genetics | Attention_Disorder | Born_an_Even_Day | Car_Accident | Fatigue | Allergy | Coughing | Output | SDP |
|---|---|---|---|---|---|---|---|---|---|---|---|---|---|
| 0 | 1 | 1 | 0 | 0 | 0 | 1 | 0 | 1 | 1 | 0 | 1 | 1 | 0.926272 |

## Local rule explanation

### 0.5 <= Smoking <= inf and 0.5 <= Coughing <= inf

Figure 7: Screenshot of a web-App showing the Sufficient Explanations and LXI of an observation chosen randomly

At the bottom of figure 7, we observe the rule associated with the first Sufficient Explanation which is $\{$**Smoking** $= True$, **Coughing** $= True\}$. Note that this rule is very powerful, because it has a coverage of $46\%$ and an accuracy of $93\%$.

**Comparison of SHAP values and LXI on LUCAS:** We can also compare the SHAP values and LXI on this dataset. In figure 8, we observe that it is the value of Coughing that is really important for this observation. Indeed it appears in several Sufficient Explanations (80%). On the other side, in figure 9, SHAP associates values to many more variables, and it is difficult to discriminate between the important values. It is difficult to deduce from the values of Smoking, Coughing, Fatigue, Allergy which is the most important variable with SHAP values.

### All sufficient explanations

|   | Sufficient explanations | SDP |
|---|---|---|
| 0 | Smoking - Coughing | 0.9387 |
| 1 | Smoking - Fatigue | 0.9078 |
| 2 | Coughing - Yellow_Fingers | 0.9225 |
| 3 | Coughing - Anxiety | 0.9036 |
| 4 | Coughing - Fatigue - Peer_Pressure | 0.9034 |

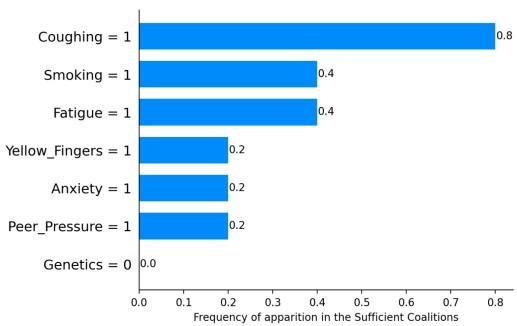

Figure 8: Sufficient Explanations and Local Explanatory Importance

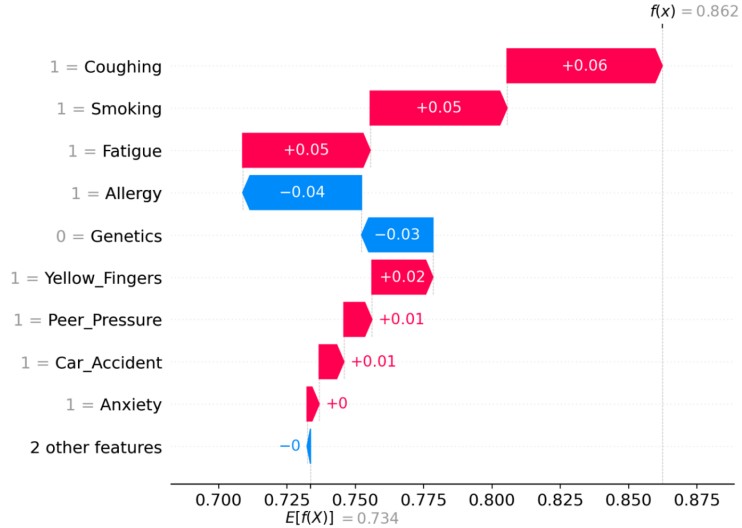

Figure 9: SHAP values

## C.3 Stability of Anchors and Sufficient Rules

Here, we run the last experiment of Section 5 on the stability of the local rules (Anchors, Sufficient Explanations) with different parameters. It consists of evaluating the stability of the methods w.r.t to input perturbations. For a given observation $x$, we compare the rule of each method with the rules obtained for 50 noisy versions of $x$ by adding random Gaussian noises $\mathcal{N}(0, \epsilon \times I)$ to the values of the features with different $\epsilon = 0.01, 0.001$.

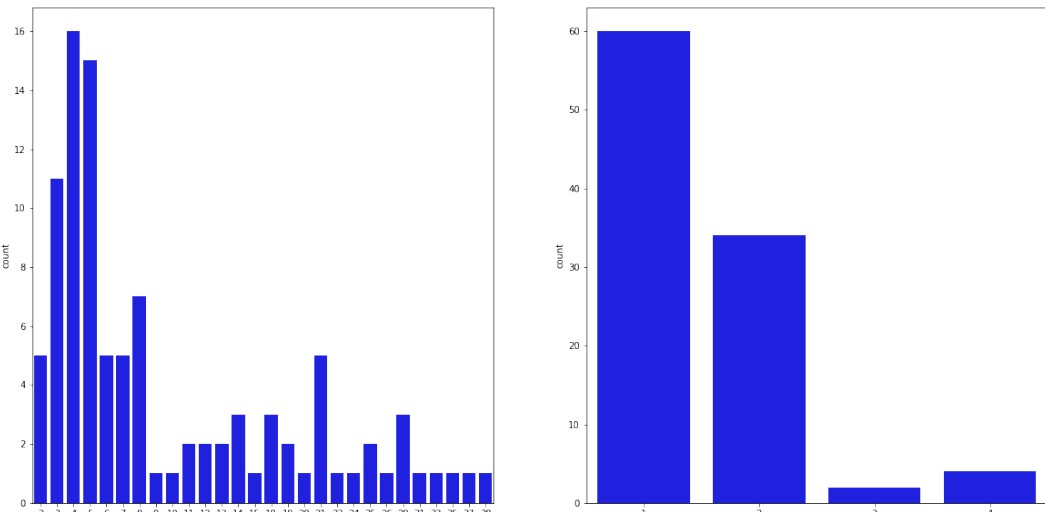

Figure 10: Size distribution of Anchors (left) and Sufficient Rules (right) when $\epsilon = 0.001$

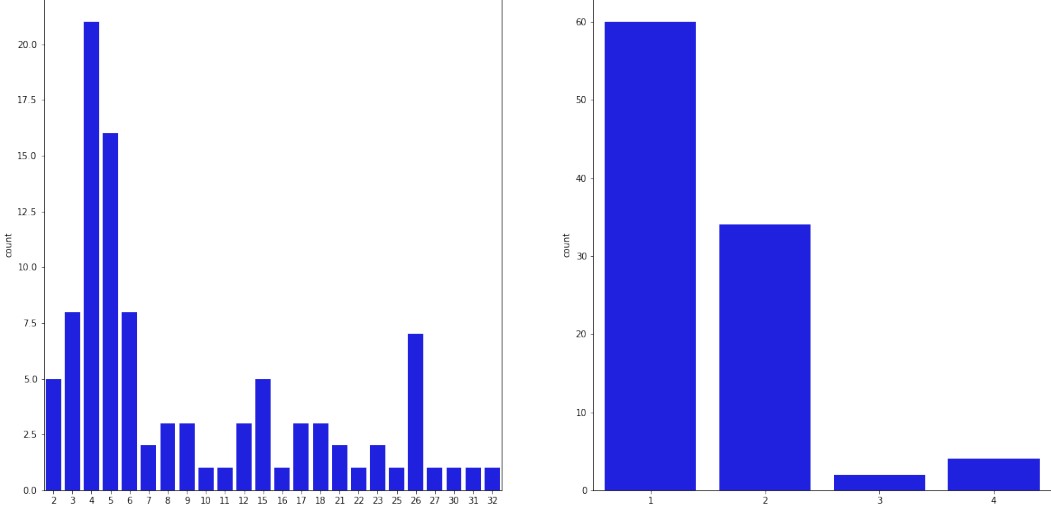

Figure 11: Size distribution of Anchors (left) and Sufficient Rules (right) when $\epsilon = 0.01$

# D From Sufficient Rules to Global Interpretable model

In this section, we investigate the capacity of transforming the Sufficient Rules explanations as a competitive global model. Indeed, we can build a global model by combining all the Sufficient Rules found for the observations in the training set. We set the output of each rule as the majority class (resp. average values) for classification (resp. regression) of the training observations that satisfy this rule. Note that some rules can overlap and an observation can satisfy multiple rules. To resolve these conflicts, we use the output of the rule with the best precision (accuracy or $R^2$). We have experimented on 2 real-world datasets: **Diabetes** [Kaggle, 2016] contains diagnostic measurements and aims to predict whether or not a patient has diabetes, **Breast Cancer Wisconsin (BCW)** [Dua and Graff, 2017] consists of predicting if a tumor is benign or not using the characteristic of the cell nuclei. Thus, we perform comparisons between the global model induced by the Sufficient Rules (G-SR) and SOTA global rule-based models as baseline. We use the package imodel [Singh et al., 2021] (resp. scikit-learn [Pedregosa et al., 2011]) for RuleFit, Skoped Rule (SkR) (resp. Decision Tree (DT), Random Forest (RF)). In table 1, we observe that the G-SR performs as well as the best baseline models while being transparent in its decision process. These experiments increase the trustworthiness of our explanations because we derive an interpretable (by-design) global model without paying a trade-off with performance. As a by-product, SR can be used as a new way of building glass-box models, but this line of research is beyond the scope of the current work.

Table 1: Accuracy of the different models on Diabetes and Breast Cancer Wisconsin dataset (BCW). For G-SR, we add the coverage of the model on the test-set in brackets.

| DATA SET | G-SR | RULEFIT | SKR | DT | RF |
|---|---|---|---|---|---|
| DIABETES | 0.98 (81%) | 0.76 | 0.71 | 0.90 | 0.92 |
| BCW | 0.95 (92%) | 0.95 | 0.93 | 0.95 | 0.96 |

These experiment permit to enforce the trust in the explanations given by our methods and open new avenues to build a glass-box model with the SR.

# E  Projected Forest CDF algorithm

---

**Algorithm 1:** Projected Forest CDF: $\widehat{F}_S$

---

1: **Inputs :** A random forest fit with $\mathcal{D}_n$, a query point $\boldsymbol{x}_S$, $y$, *min_nodes_size*
2: **Output :** $\widehat{F}(y|\boldsymbol{X}_S = \boldsymbol{x}_S)$
3: **for** all trees in the forest **do**
4:      initialize *nodes_level* as a list of nodes containing only the root node;
5:      initialize *nodes_child* as an empty list of child nodes;
6:      initialize *samples* as the list of observation indices of the full training data of the tree;
7:      **for** all levels in the tree **do**
8:        **for** all nodes in *nodes_level*: **do**
9:          **if** the node splits on a variable in $S$ **then**
10:            compute whether $\boldsymbol{x}_S$ falls in the left or right child node;
11:            append the child node to *nodes_child*;
12:            set *samples_child* as the observations in *samples* which satisfy the split;
13:          **else**
14:            append both the left and right children nodes to *nodes_child*;
15:            set *samples = samples_child*;
16:          **end if**
17:          **if** the size of *samples_child* is lower than *min_node_size* **then**
18:            break the loop through the tree levels;
19:          **else**
20:            set *samples = samples_child*;
21:          **end if**
22:        **end for**
23:        set *nodes_level = nodes_child*;
24:      **end for**
25:      compute the tree prediction as the average of $\mathbb{1}_{Y_i \leq y}$ for all $i$ in *samples*;
26: **end for**
27: **return** average the prediction of all trees;

---