# OpenReview forum: "Consistent Sufficient Explanations and Minimal Local Rules for explaining the decision of any classifier or regressor"
_NeurIPS.cc/2022/Conference — NeurIPS 2022 Accept_

### Official Review · Reviewer_bHU5 · 2022-07-07

**Rating:** 6
**Confidence:** 4
**Soundness:** 3 good
**Presentation:** 3 good
**Contribution:** 3 good

**Summary:**

This paper proposes a method for finding important features as Consistent Sufficient Explanation.
The essential building block of the method is the conditional probability estimation with conditioning on arbitrary subset of the features.
The authors proposed using Quantile Regression Forest and Projected Forest to estimate such conditional probability.
The authors also proved that the estimated conditional probability converges to the true probability as the number of samples goes to infinite.
The experimental results show that the proposed estimator provides more accurate estimate of important features than the existing methods such as LIME, SHAP, INVASE, and Anchors.

**Questions:**

Are there any other possible approaches for estimating the conditional probability estimation with conditioning on arbitrary subset of the features?

**Limitations:**

The authors mentioned that the proposed method is suitable for tabular data.

**Strengths And Weaknesses:**

## Strength
### Originality
The notion of Consistent Sufficient Explanation was proposed in a previous study.
The major novelty of the paper is therefore on constructing an estimator of the conditional probability with conditioning on arbitrary subset of the features.
This will be a topic that is less explored in the literature.
The proposed estimator based on Quantile Regression Forest and Projected Forest seems reasonable, particularly for tabular data.

### Clarity, Quality
Both the notion of Consistent Sufficient Explanation and its extensions as well as the proposed estimator are described clearly.

### Significance
The results on the experiments with synthetic data and a few real world datasets looks promising.

## Weakness
*(My comment here is rather suggestion than Weakness.)*
In the paper, the authors focused on the conditional probability estimation based on random forest.
I wonder if there is any other type of approaches we can think of.
Discussing possible alternative approaches would be beneficial to the readers.
One naive approach would be to train a generative model $p(X_{-S} | X_S)$, e.g., by using neural networks [Ref1,2,3,].
We can then estimate the conditional probability $p(Y | X_S) = \int p(Y | X_S, X_{-S}) p(X_{-S} | X_S) d X_{-S}$, e.g., by sampling $X_{-S}$ from the generative model.
Although this is apparently computationally expensive, this can be an alternative approach when the tree-based models are not appropriate such as images.

[Ref1] Variational Autoencoder with Arbitrary Conditioning
[Ref2] ACFlow: Flow Models for Arbitrary Conditional Likelihoods
[Ref3] Arbitrary Conditional Distributions with Energy

---

> ### Author Response · Authors · 2022-08-02
> **Response reviewer bHU5**
>
> We thank the reviewer for his detailed analysis and suggestions. We start with a complimentary note about the summary.
>
> > C:"This paper proposes a method for finding important features as Consistent Sufficient Explanation. The essential building block of the method is ....The experimental results show that the proposed estimator provides more accurate estimate of important features than the existing methods such as LIME, SHAP, INVASE, and Anchors."
>
>
> We would like to emphasize that our work is not only dedicated to idendifying local important variables; one of our main contributions is that we produce a "sparse" rule (based on local important variable) that can be used for explaining/predicting with theoretical guarrantees.
>
> > C: The results on the experiments with synthetic data and a few real world datasets looks promising.
>
> We provide more experiments in the Appendix.
>
> > Q: In the paper, the authors focused on the conditional probability estimation based on random forest. I wonder if there is any other type of approaches we can think of. Discussing possible alternative approaches would be beneficial to the readers. One naive approach would be to train a generative model , e.g., by using neural networks.
>
> We agree with the suggestion that alternative ways of computing the conditional probability $p(X_S|X_{\bar{S}})$ (with the suggested readings could be a promising research direction). At the time, we are focused mainly on tabular data. The explainability needs for images, NLP,... are specific and well-addressed by Neural Networks. Nevertheless, we have tried the suggested approach based on Neural Networks for learning conditional distributions, such as conditional tabGan [Modeling Tabular Data using Conditional GAN (2019)], but we found that it was computationally costly and difficult to calibrate, and we did not go further. This motivated us to consider Tree-based approximations that provide (relatively fast) algorithms and for which we can have some theoretical guarantees, while it is not yet the case for conditional distributions learned with NN. In addition, trees are easy to use and calibrate.

---

> > ### Comment · Reviewer_bHU5 · 2022-08-08
> > **Reply**
> >
> > I would like to thank the authors for the clarification and reply to my comment.

---

### Official Review · Reviewer_aVcg · 2022-07-11

**Rating:** 5
**Confidence:** 1
**Soundness:** 3 good
**Presentation:** 1 poor
**Contribution:** 3 good

**Summary:**

The paper proposes LXI, a feature importance metric, and Minimal Sufficient Rule, a generalization of Minimal  Sufficient Explanation.
 - A Minimal Sufficient Explanation (M-SE) of an instance in a model is the instance's smallest subset of features such that, w/ high probability, the model's prediction using the subset is close to the prediction when using all features.
 - LXI defines the important features of an instance in a model to be the subset of features that frequently appears in the instance's M-SE.
 - A Minimal Sufficient Rule of an instance in a model is the maximal bounding box around a subset of the instance's feature values such that every M-SE to the instance is also a SE to every point inside the box.

The paper proposed to construct a Minimal Sufficient Rule using projected forest and quantile regression forest.

**Questions:**

I'd appreciate if the authors can confirm or reject my understanding of the various definitions in my summary.

**Limitations:**

Yes

**Strengths And Weaknesses:**

Given my understanding of the definition of Minimal Sufficient Rule, I don't think I can make assertive judgment about the paper: As noted in the paper and the Anchor paper, a "rule" is a set of if-then predicates. How can a set of bounding boxes be interpreted as such?

The main contribution of the paper, the construction of a Minimal Sufficient Rule, is explained in only 10 lines (line 294-304). Perhaps a longer explanation together with some examples can help me better understand the the paper.

What is the purpose of LXI? It seems disconnected to the rest of the paper. As a feature importance metric, it is not compared to other feature importance metric in the literature, nor is it used in the proposed method.

The paper claims to "introduce" Minimal Sufficient Explanations, but this is not true. It has been extensively studied in some of the references, e.g. Wang et al 2021. Perhaps this paper is the first to extend the definition to regression?

---

> ### Author Response · Authors · 2022-08-02
> **Response reviewer aVcg**
>
> We thank the reviewer for the comments and questions raised.
> > Q: Commment: "Given my understanding of the definition of Minimal Sufficient Rule, I don't think I can make assertive judgment about the paper: As noted in the paper and the Anchor paper, a "rule" is a set of if-then predicates. How can a set of bounding boxes be interpreted as such?"
>
> The "IF-THEN" interpretation is motivated by a decision tree. A decision tree makes a partition of the feature space $\mathbb{R}^p$ into rectangles/boxes, where the final decision/prediction is made when we are in the leaf/box. Conversely, a box/rectangle $\prod_{i=1..p} [a_i,b_i]$ can be then seen as the leaf of a decision tree, with a sequence of decision for $x$,  $IF \; a_1 \leq x_1 \leq b_1 \quad AND \quad \dots \quad   AND \quad  a_p \leq x_p \leq b_p  \quad THEN f(x) = 0$.
>
> > Q: The main contribution of the paper, the construction of a Minimal Sufficient Rule, is
> explained in only 10 lines (line 294-304). Perhaps a longer explanation together with some examples can help me better understand the the pape
>
> The paper's main contributions are the Same Decision Probability (its extension to regression,
> its computation, and its consistency) and the use of SDP for building Minimal Sufficient Rules. The rules
> are built on top of the SDP, such that they can be turned into an easy-to-interpret distilled model.
> Consequently, lines 294-304 are a wrap-up of the previous results on SDP and of the partitions derived
> from the Projected Random Forests. We have provided in Appendix C a detailed construction of the
> Local Rules with examples on the Moon Dataset and LUCAS (bayesian causal dataset). However, we will give more details about how the leaves of the random Forest are merged to get the Sufficient Rules.
>
> > Q: What is the purpose of LXI? It seems disconnected to the rest of the paper. As a
> feature importance metric, it is not compared to other feature importance metric in the literature, nor is it used in the proposed method
>
> LXI is not one of the main contributions of the paper, but it is a summary of the Minimal Sufficient
> Explanations: indeed, for each observation, we can have several Sufficient Explanations, and LXI is the
> frequency of a variable in Minimal Sufficient Explanation/Rule. Hence, the stability of a variable $X_i$
> can reveal the importance of a variable. This is similar to the importance measure in trees/random forests,
> where the number of nodes where a given variable was chosen is used as an importance score. The relevancy of the LXI index is show-cased in the Appendix
> C: on the Moon DataSet and the LUCAS dataset (bayesian causal dataset) where LXI exhibits sharper results than SHAP, closer to the Ground Truth
>
>
> > Q: The paper claims to ”introduce” Minimal Sufficient Explanations, but this is not true.
> It has been extensively studied in some of the references, e.g. Wang et al 2021. Perhaps this paper is the first to extend the definition to regression?
>
> We refer to the paper of Wang et al. 2021 for introducing Probabilistic Sufficient Explanation,
> see line 36 (and line 41 for related works). In section 3, we clearly acknowledge the prior introduction
> (line 100, Chen et al) of SDP. As the reviewer remarks it, we claim that we extended SDP and
> P-SE to regression, and we propose a guaranteed estimator of SDP (while before, it was required to
> have discrete variables and to estimate the full distribution of X). Nevertheless, we claim that we have
> introduced the Minimal Sufficient Rules, and the Local eXplanatory Importance index. We will rephrase line 403 in our conclusion in order to avoid any confusion.
>
> >Q: I’d appreciate if the authors can confirm or reject my understanding of the various definitions in my summary.
>
> We agree with your summary.

---

> > ### Comment · Reviewer_aVcg · 2022-08-07
> > **Thank you for your answers**
> >
> > Thank you for your answers. You addressed all of my questions. I have raised my score to 5.

---

### Official Review · Reviewer_HAsx · 2022-07-11

**Rating:** 7
**Confidence:** 3
**Soundness:** 3 good
**Presentation:** 3 good
**Contribution:** 3 good

**Summary:**

This paper focuses on the problem of explaining the model output by extending the notion of probabilistic sufficient explanations (P-SE) to regression problems and non-binary settings. To that end, the authors extend the notion of same decision probability (SDP) to regression tasks and propose an estimator for it based on ideas from projected forest and quantile regression forest. Finally, they also introduce a local-rule based explanation, namely minimal sufficient rules, by generalizing sufficient explanations.

**Questions:**

How does the local explanatory importance compare, for example to SHAP values, on the tasks discussed in the experiments section? I am also unsure about its value as local feature attribution in practical settings, as it requires computing the A-SE or M-SE first which is expensive. (Unless there’s an efficient algorithm to compute/estimate it, which I could not find in the paper).

I would encourage the authors to also evaluate the proposed explanations on real-world benchmark datasets for regression tasks. In the synthetic model, most features are noise variables which is not what most real-world data or models would look like.

Table 1: Is correctness the same as consistency described in the previous paragraph (end of page 8)?

Lines 359-360: typo with comma and decimal points

**Limitations:**

Yes

**Strengths And Weaknesses:**

(+) The paper proposes several local explanation methods to address an interesting problem of explaining regression models.

(+) A fast estimator for the SDP is also a novel and significant contribution. A theoretical analysis of its convergence is provided as well.

(+) The paper is overall well written, and related works are discussed clearly.

(-) For empirical evaluation on regression tasks, only a toy example is considered.

(-) Local explanatory importance was introduced but not discussed or evaluated further.

---

> ### Author Response · Authors · 2022-08-02
> **Response reviewer HAsx**
>
> We really appreciate that you acknowledge our contributions and find our ideas "well written" and "novel and significant contributions". We appreciate your specific suggestions, which will help us improve our paper's quality. We thank you for the time you took to review the paper.
>
> > Q: Local explanatory importance was introduced but not discussed or evaluated further.
>
> We have noticed in Lines 350-351 that we provide comparisons of LXI and SHAP in the Appendix. We provide a comparison of two datasets, a semi synthetic dataset (LUCAS) and a toy dataset (moon dataset). Indeed, we could not put them in the paper because of lack of space, but we will put them in the final version.
>
>
> > Q: How does the local explanatory importance compare, for example to SHAP values, on the tasks discussed in the experiments section? I am also unsure about its value as local feature attribution in practical settings, as it requires computing the A-SE or M-SE first which is expensive. (Unless there's an efficient algorithm to compute/estimate it, which I could not find in the paper).
>
> As said above, we provide a comparison of SHAP and LXI on two examples in the Appendix. We will refer to this comparison in the paper when we introduce LXI. The Local Explanatory Importance is a by-product of the estimation of A-SE and M-SE  as it summarizes the frequency of a given variable in Minimal Sufficient Explanations. Hence the crux is the computation of the M-SE: for this, we use the Projected Random Forest algorithm that provides a very efficient way to compute the SDP. In addition, we restrict the search of M-SE among a relatively small number $k<<p$ (we use $k=10$).
>
> > Q: I would encourage the authors to also evaluate the proposed explanations on real-world benchmark datasets for regression tasks. In the synthetic model, most features are noise variables which is not what most real-world data or models would look like.
>
> We agree with the suggestion, but note that the evaluations on real-world datasets is very difficult since we do not known the ground truth (the truth important variables or rules). Therefore, we propose to evaluate the predictive capacity of the Sufficient Rules explanations instead. Indeed, we can build a global model by combining all the Sufficient Rules found for the observations in the training set. We set the output of each rule as the majority class(resp. average values) for classification (resp. regression) of the training observations that satisfy this rule. Note that some rules can overlap and observation can satisfy multiple rules. To resolve these conflicts, we use the output of the rule with the best precision (accuracy or R2). We have experimented on 2 real-world regression datasets: Bike [kaggle, 2014] and Diabetes [UCI]. The RF has a mean absolute error of 15.7, 32.18, and the Sufficient Rules as a predictor (G-SR) have 16.91, 54.68 for Bike and Diabetes, respectively. We observe that the G-SR performs as well as the baseline models while being transparent in its decision process. These experiments increase the trustworthiness of our explanations because we derive an interpretable (by-design) global model without paying a trade-off with performance. As a by-product, SR can be used as a new way of building glass-box models, but this line of research is beyond the scope of the current work. We will add some examples on real-world regression datasets in the final version.
>
> > Q:Is correctness the same as consistency described in the previous paragraph (end of page 8)?
>
> Yes, we apologize for this typo. We will replace consistency with correctness in the final version.

---

> > ### Comment · Reviewer_HAsx · 2022-08-08
> > **Response**
> >
> > Thank you for your answer and clarifications.

---

### Official Review · Reviewer_rsCo · 2022-07-13

**Rating:** 4
**Confidence:** 4
**Soundness:** 2 fair
**Presentation:** 2 fair
**Contribution:** 2 fair

**Summary:**

In this paper, the authors propose a framework that can find sufficient explanations for any model of the form $x,f(x)$ and don't explicitly require access to the black box model $f$. One of their key claims to is to propose the first method to provide local rule-based explanations for the regression problem. They leverage prior art on Same Decision Probability (SDP) and in order to obtain a consistent SDP estimator to obtain MSE's they build upon ideas from Quantile Regression Forests and Projected Forests. Certain heuristics are used to alleviate the computational burden which are also inspired from more recent prior art. Empirical results include comparison against rule-based methods such as Anchors, etc to indicate better performance.

**Questions:**

1) How would you justify choices of certain parameters/heuristics as mentioned above ?

2) What notion of stability are you presenting here (line 387). There are several notions of stability within the XAI literature hence mathematical definition or citation should  be provided ? For example is it the stability defined from here (Alvarez-Melis David, Jaakkola Tommi S. On the robustness of interpretability methods -  ICML Workshop on Human Interpretability in Machine Learning. 2018.)

3) I do not see any citation or comparison w.r.t Contrastive Explanations techniques (https://arxiv.org/pdf/1802.07623.pdf) ? These have been applied successfully to tabular data also and seem very relevant work to compare against? https://arxiv.org/abs/1906.00117




**Ethics Review Area:**

["I don’t know"]

**Limitations:**

Yes

**Strengths And Weaknesses:**

**Strengths**

1) Authors try to address a relatively unaddressed problem on local rule-based explanations for regression problems.
2) Empirical results and overall writeup are coherent with the theme the authors wanted to convey.

**Weakness**

1) Choice on certain heuristics like choosing 10 (line 290) seem to be directly used from prior art with no justification.
2) Limited scope of applicability to tabular datsets only

---

> ### Author Response · Authors · 2022-08-02
> **Response reviewer rsCo**
>
> > Q: How would you justify choices of certain parameters/heuristics as mentioned above ?
>
> The main hyperparameters are:
> - $k:$ number of preselected variables
> - $\pi:$ the minimal probability of changing the decision,
> And for regression problems, there is
> -  $t:$ the radius of the ball center at the prediction
>
> The choice of $k$ << $p$ (number of variables) is motivated by the fact that most datasets have intrinsic dimensions much lower than the ambient dimension. As we noticed in Lines 291-293, our Selection criterion is based on Proposition 1 in [Scornet et al. 2015], which highlights the fact that RF naturally splits the most on influential variables. The choice of $k=10$ is directly driven by our computation power required to explore the $2^{10}$ subsets. It is, of course, possible to increase $k$ or/and to change the way we select the k variables, but we remark that we have always found coalitions S with a probability above $\pi=0.9$ for most datasets. If it is not possible, then we suggest increasing k until we can find coalitions and minimal sufficient explanations for every observation.
>
> We suggest choosing $\pi = 0.9$ as it is an acceptable level of risk, but the user can increase this probability depending on the use case.
>
> Probably the newest and most challenging hyperparameter to select is $t$; we suggest having an adaptive $t$ being equal to the $1-\alpha$ quantile of the conditional distributions $Y \vert \boldsymbol{X} = \boldsymbol{x}$ (obtained as a by-product of the computation of the SDP bu using the Random Forest as Quantile Regression Forest [Meinshausen and Ridgeway, 2006]): we build then a confidence interval of varying length but constant confidence level across the dataset. This makes our approach as a natural generalization of SDP in the classification case by accounting for the uncertainty of the model to explain $f$. In that case, the SDP and associated coalitions $S$ should be read: "if $\boldsymbol{X} = \boldsymbol{x}$ is fixed, then there is a probability at least $\pi$ of not changing the prediction significantly, with level $1-\alpha$."
> For this reason, we suggest fixing $\alpha$ and $\pi$ at standard level $1-\alpha = \pi = 0.9$ agreeing with acceptable level of risks.
>
>
> > Q: What notion of stability are you presenting here (line 387). There are several notions of stability within the XAI literature; hence mathematical definition or citation should be provided ? For example is it the stability defined from here (Alvarez-Melis David, Jaakkola Tommi S. On the robustness of interpretability methods - ICML Workshop on Human Interpretability in Machine Learning. 2018.)
>
> We are interested in input perturbations, i.e., testing whether nearby observations return the same explanations as in [Alvarez-Melis David, Jaakkola Tommi S. On the robustness of interpretability methods - ICML Workshop on Human Interpretability in Machine Learning. 2018.]. However, we do not compute their Lipschitz constant but the number of different rules after perturbations.
>
>
> > Q: I do not see any citation or comparison w.r.t Contrastive Explanations techniques (https://arxiv.org/pdf/1802.07623.pdf) ? These have been applied successfully to tabular data also and seem very relevant work to compare against. https://arxiv.org/abs/1906.00117
>
> We thank the reviewer for pointing out these two papers: we think that they both address a close but dual problem. Our focus is on identifying the group
> of variables that can *keep the decision* while in the papers, the objective is to generate counterfactuals that can *change the prediction*. But we can generalize our approach to solve the dual problem, i.e., find the variables that have a good probability of changing the decision. We can compute the probability of changing the decision for a coalition of variable S with the Random Forest as the SDP, and identify the minimal subset for changing the decision. Indeed, we have already developed these ideas in another paper. As the current paper is already quite long, we propose to discuss these ideas briefly in the Appendix.
>
> > C: Technically solid paper where reasons to reject, e.g., limited evaluation, outweigh reasons to accept, e.g., good evaluation. Please use sparingly.
>
> Note that we conducted many experiments in the Appendix.

---

> > ### Comment · Reviewer_rsCo · 2022-08-08
> > **Response on author rebuttal**
> >
> > I appreciate the authors taking time and effort to clarify my concerns with the paper. I am satisifed with the response on the choice of parameters. However, I do not completely agree on the differences authors have pointed out w.r.t the prior art I had shared above. ``Our focus is on identifying the group of variables that can keep the decision while in the papers, the objective is to generate counterfactuals that can change the prediction.``  - Request authors to clearly read through notion of **pertinent positives** in the paper above and you would notice the notion of preserving class prediction also within a PP which is very similar to your notion. In light of this, I would like to keep my original score.

---

> > > ### Author Response · Authors · 2022-08-09
> > > **Brief comment about the link between PP and sufficient explanation**
> > >
> > > We agree with reviewer rsCo that the concept of Pertinent Positive developed is related to the concept of sufficient minimal explanation.
> > > The Pertinent Positive is defined as “a factor whose presence is minimally sufficient in justifying the final classification” , it is defined mathematically by an optimisation problem, similar to the optimisation problem used for finding counterfactuals. A complementary definition is also “a PP for an input x as the sparsest example (w.r.t. base values) whose feature values are no farther from the base values than those of x, with it lying in the same class as x.”
> > > We acknowledge that the concept of PP can be phrased literally in a way similar than  Sufficient Minimal Explanation and that the two cited papers are not completely dedicated to finding counterfactuals.
> > > But the Contrastive Explanation gives a complete explanation with necessary elements for changing or keeping the decision, in the spirit of counterfactuals, and the PP concept appears to be introduced and discussed as a mirroring concept of Pertinent Negatives, and not for itself.
> > > In addition, the construction of PP stays largely inspired by the mathematical definition of Counterfactuals, which is definitely different from our approach based on Same Decision Probability. The approach with PP is also focused on classification, while our approach can deal with classification and regression. The mathematical technics and results we obtain are not related and cannot be extended straightforwardly to the approach of PP and Contrastive Explanations.
> > >
> > > We will refer to the papers on PP in order to show that similar principles and attempts are conducted for selecting small groups of local important and interpretable variables. But, we
> > > also put emphasis on the fact that the papers introducing PP and Contrastive Explanations and the papers about SDP and Sufficient Explanations (Wang 2020, Shih 2018) have not been related before, despite this similarity and the fact that these works on SDP originates from series of papers about Bayesian networks dating back to 2003, see [Chan and Darwiche, 2003] Hei Chan and Adnan Darwiche. Rea-
> > > soning about Bayesian network classifiers, UAI.
> > > While our work is clearly a generalisation and development of this anterior works on SDP, we still think that the PP concept is not directly relevant to our main contributions, and that a valuable discussion of the link between PP and SDP and sufficient Minimal Explanations would need a dedicated thorough work that goes beyond this paper (who already offers several significant contributions and comparisons with mainstream methods).

---

### Comment · Area_Chair_ktzA · 2022-08-03
**Please start author-reviewer discussion**

Hi authors and reviewers,

The discussion phase has begun. Please read the other reviews and the author's response (if the authors choose to submit one) and start discussing them with the other reviewers, the authors, and myself.

Note that by default, the authors can see the discussions posted by the reviewers (and vice versa). Please use the "Readers" field to adjust the audience of your post if so wished.

*Our goal is to contribute to the discussion to reach a consensus on each paper*. There is only one week for the discussion (until **August 9**). So please do not wait and start the discussion immediately. Thank you very much.

Best,\
The AC

---

### Meta-Review · Area_Chair_ktzA · 2022-08-26

**Recommendation:** Accept
**Confidence:** Less certain

**Metareview:**

I have read all comments and responses carefully.

Reviewers praise the novelty of the solution and recognized the under-explored nature of the problem. The proposed estimator seems reasonable, particularly for tabular data. Reviewers complained about the lack of explanation on some heuristics used and the limited scope of applicability of the method.

Overall, reviewers agree that this is an important and yet underexplored problem and the authors have provided useful contributions. I, therefore, have decided to recommend the acceptance of the paper.

**Award:**

No

---

### Decision · Program_Chairs · 2022-09-14

Accept